# Robust Testing in High-Dimensional Sparse Models

**Anand Jerry George**
School of Computer and Communication Sciences
École Polytechnique Fédérale de Lausanne (EPFL)
`anand.george@epfl.ch`

**Clément L. Canonne**
School of Computer Science
The University of Sydney
`clement.canonne@sydney.edu.au`

## Abstract

We consider the problem of robustly testing the norm of a high-dimensional sparse signal vector under two different observation models. In the first model, we are given $n$ i.i.d. samples from the distribution $\mathcal{N}(\theta, I_d)$ (with unknown $\theta$), of which a small fraction has been arbitrarily corrupted. Under the promise that $\|\theta\|_0 \leq s$, we want to correctly distinguish whether $\|\theta\|_2 = 0$ or $\|\theta\|_2 > \gamma$, for some input parameter $\gamma > 0$. We show that any algorithm for this task requires $n = \Omega\left(s \log \frac{ed}{s}\right)$ samples, which is tight up to logarithmic factors. We also extend our results to other common notions of sparsity, namely, $\|\theta\|_q \leq s$ for any $0 < q < 2$. In the second observation model that we consider, the data is generated according to a sparse linear regression model, where the covariates are i.i.d. Gaussian and the regression coefficient (signal) is known to be $s$-sparse. Here too we assume that an $\varepsilon$-fraction of the data is arbitrarily corrupted. We show that any algorithm that reliably tests the norm of the regression coefficient requires at least $n = \Omega\left(\min(s \log d, 1/\gamma^4)\right)$ samples. Our results show that the complexity of testing in these two settings significantly increases under robustness constraints. This is in line with the recent observations made in robust mean testing and robust covariance testing.

## 1 Introduction

Hypothesis testing is a fundamental task in statistics and a staple of the scientific method, in which we seek to test the validity of a pre-specified hypothesis based on empirical observations. In this work, we are concerned with the problem of testing whether a given high-dimensional sparse signal vector is zero under two common and well-studied observation models: 1) the Gaussian location model and 2) the Gaussian linear regression model. Specifically, in the Gaussian location model, we observe i.i.d. samples from a $d$-dimensional spherical Gaussian distribution with an unknown sparse mean vector, and seek to detect whether its $\ell_2$ norm is large (equivalently, we observe a set of measurements subject to white noise, and seek to determine whether there exists an underlying (sparse) signal). Similarly, in the Gaussian linear regression model we seek to detect whether the $\ell_2$ norm of the sparse regression coefficient is large. We further assume that our samples are imperfect or even corrupted, allowing an adversary to arbitrarily tamper with up to an $\varepsilon$-fraction of the observations. Our objective is to characterize the minimum number of samples required to perform these testing tasks, and, crucially, to understand the effect that requiring robustness to this adversarial corruption has on the complexity of the problems.

It is known [DKS17, DK21] that, for a variety of high-dimensional tasks, robust testing becomes as costly (in terms of sample complexity) as the corresponding *estimation* task. This is in contrast to the non-robust version, where testing is typically much more efficient – often by a quadratic factor in the dimension. However, it is unclear how sparsity enters the picture, and for instance if robustness only starts becoming "costly" when the signal vector is sufficiently dense – i.e., whether the problem

36th Conference on Neural Information Processing Systems (NeurIPS 2022).

exhibits a phase transition. This is particularly relevant, as the non-robust versions of the problems we consider are known to present such a phase transition at sparsity $s \approx \sqrt{d}$.

> *How does robustness affect the sample and computational complexities of testing norm of the signal vector in high-dimensional sparse models? Does testing remain easier than learning?*

This type of question, framed in a minimax setting, sits at the intersection of theoretical computer science (where it is captured under the framework of distribution testing) and robust statistics. Specifically, for $\Theta \subseteq \mathbb{R}^d$, let $\{\mathbf{p}_\theta\}_{\theta \in \Theta}$ be a family of distributions and $\mathbf{p}_0$ be a reference distribution (simple null hypothesis–in our case the standard Gaussian). Then, we say that an algorithm $T$ reliably tests the $\ell_2$-norm of $\theta$ if it satisfies the condition

$$\max \left\{ \Pr_{X \sim \mathbf{P}_0^n}[T(X) = \mathsf{reject}], \sup_{\substack{\theta \in \Theta \\ \|\theta\|_2 \geq \gamma}} \Pr_{X \sim \mathbf{P}_\theta^n}[T(X) = \mathsf{accept}] \right\} \leq \delta \tag{1}$$

where $\delta \in (0, 1]$ is the failure probability, which following the literature we will hereafter set to $1/3$.[1] The quantity of interest here is the *sample complexity*, that is the minimum number of samples $n$ required by any algorithm to solve the problem.

The above task, however, assumes access to "perfect" samples from the unknown distribution $\mathbf{p}_\theta$. This is often an unrealistic assumption, as a fraction of the $n$ samples could be imperfect or corrupted. This motivates the setting of *robust* testing. The problem is then similar to the formulation in (1), with a crucial difference: the algorithm $T$ does not have access to the i.i.d. samples $X = (X_1, \ldots, X_n) \in \mathbb{R}^{d \times n}$, but instead to a "contaminated" version $\tilde{X} = (\tilde{X}_1, \ldots, \tilde{X}_n) \in \mathbb{R}^{d \times n}$ obtained by arbitrarily modifying up to $\varepsilon n$ of the $X_i$'s (i.e., an $\varepsilon$-fraction). We will refer to this as the $\varepsilon$-*corruption model*.

In this work we consider two instances of the general testing task (1) in the robust testing setting. First, let us introduce some notation common to the problems. For $q \in (0, 2)$ let $\mathcal{B}_{s,q} := \{\theta \in \mathbb{R}^d : \|\theta\|_q \leq s\}$ be the $\ell_q$−ball of radius $s$ in $\mathbb{R}^d$. For $q = 0$, we get the usual notion of sparsity, and will simply write $\mathcal{B}_s$ for $0 \leq s \leq d$. Let $\mathcal{N}(\theta, I_d)$ denote the $d$-dimensional Gaussian with mean $\theta \in \mathbb{R}^d$ and identity covariance.

**Sparse Gaussian mean testing.** The first problem that we consider is the *sparse Gaussian mean testing*, in which, given an $\varepsilon$-corrupted dataset of $n$ samples from $\mathcal{N}(\theta, I_d)$, where $\theta \in \mathcal{B}_{s,q}$ is unknown, our goal is to robustly distinguish between (1) $\|\theta\|_2 = 0$ , and (2) $\|\theta\|_2 \geq \gamma$ (equivalently, the total variation distance between $\mathcal{N}(\theta, I_d)$ and the standard Gaussian $\mathcal{N}(0, I_d)$ is $\Omega(\gamma)$), for some input parameter $\gamma \in (0, 1]$.

**Testing in sparse linear regression model.** This is the second problem that we consider. In the sparse linear regression model the data is generated according to the following process: Let $X_1, X_2, \cdots, X_n$ be i.i.d. samples from $\mathcal{N}(0, I_d)$. Let $\theta \in \mathcal{B}_s$ be unknown and let $\xi_i$'s be i.i.d. samples from $\mathcal{N}(0, 1)$ (and independent from the $X_i$'s), for $1 \leq i \leq n$. Then, the $y_i$'s are generated as follows:

$$y_i = \langle X_i, \theta \rangle + \xi_i \quad \text{for all } 1 \leq i \leq n. \tag{2}$$

Note that for a given $\theta$, the joint distribution of $(X_i, y_i)$ is $\mathcal{N}(0, \Sigma_\theta)$, where $\Sigma_\theta = \begin{bmatrix} I_d & \theta \\ \theta^T & 1 + \|\theta\|^2 \end{bmatrix}$.

Our aim is to robustly distinguish between (1) $\|\theta\|_2 = 0$ , and (2) $\|\theta\|_2 \geq \gamma$, given an $\varepsilon$-corrupted version of the observations $(X_1, y_1), (X_2, y_2), \cdots, (X_n, y_n)$.

Note that for both the problems, one can restrict themselves to the case $\gamma \geq \varepsilon$, as otherwise the problem becomes trivially information-theoretically impossible.

## 1.1 Our contributions

Our main contributions are the characterization of the sample complexity of robust sparse Gaussian mean testing for a range of notions of sparsity, and a lower bound on the sample complexity of robust

---

[1]The choice of the value 1/3 here is arbitrary, and any fixed value greater than 1/2 would suffice, as one can amplify the success probability to $1 - \delta$, for any $\delta > 0$, using a standard majority vote.

testing in the sparse linear regression model. Together, these results fully answer the above question, and provide more evidence to the belief that "robustness requirements make the testing tasks as hard as the corresponding estimation tasks." To establish our lower bounds, we draw upon and combine a variety of methods from the literature, in order to upper bound the $\chi^2$-divergence between a point and a mixture distribution before concluding by Le Cam's two-point method. We elaborate further on those aspects below.

**Sparse Gaussian mean testing.** It is known [CCT17] that, in the *non-robust* setting described in (1), the sample complexity of sparse Gaussian mean testing is

$$n_0(s, d, \gamma) = \begin{cases} \Theta\left(\frac{s}{\gamma^2} \log\left(1 + \frac{d}{s^2}\right)\right) & \text{if } s < \sqrt{d} \\ \Theta\left(\frac{\sqrt{d}}{\gamma^2}\right) & \text{if } s \geq \sqrt{d} \end{cases}$$

for $q = 0$, and, for $q \in (0, 2)$,

$$n_q(s, d, \gamma) = \begin{cases} \Theta\left(\frac{m}{\gamma^2} \log\left(1 + \frac{d}{m^2}\right)\right) & \text{if } m < \sqrt{d} \\ \Theta\left(\frac{\sqrt{d}}{\gamma^2}\right) & \text{if } m \geq \sqrt{d} \end{cases},$$

where $m := \max\{u \in [d] : \gamma^2 u^{\frac{2}{q}-1} \leq s^2\}$ is the *effective sparsity*. In particular, both sample complexities present a phase transition at $\sqrt{d}$, after which the sparsity no longer helps decreasing the sample complexity of the problem, which defaults to the "folklore" non-sparse bound of $\Theta(\sqrt{d}/\gamma^2)$.

Our main result in this setting is a lower bound on *robust* sparse mean testing, which shows a significantly different landscape:

**Theorem 1** (Informal; see Theorems 4 and 5). *For every constant $\varepsilon, \gamma$, the sample complexity of robust sparse Gaussian mean testing in the $\varepsilon$-corruption model is*

$$\Omega\left(s \log \frac{ed}{s}\right)$$

*for $q = 0$, and $\Omega\left(m \log \frac{ed}{m}\right)$ for $q \in (0, 2)$, where $m = \max\{u \in [d] : \gamma^2 u^{\frac{2}{q}-1} \leq s^2\}$.*

Moreover, our bound for standard $s$-sparsity is tight, in view of the known $O\left(\frac{s}{\gamma^2} \log \frac{ed}{s}\right)$ sample complexity bound for robust sparse mean *estimation* [Li17, DK19] (which implies the same bound for robust testing). This not only shows that the robust testing problem is much harder than its non-robust counterpart especially in the dense regime (and actually as hard as the robust *estimation* problem), but also that the robust setting no longer presents any threshold phenomenon. If we set $s = d$, we further recover the result in [DKS17] that robustness requirement increases the sample complexity of Gaussian mean testing. Figure 1 illustrates the sample complexity of robust and non-robust sparse Gaussian mean testing as a function of the sparsity.

The tightness of our bound, however, only follows from previous work in the case $q = 0$ (standard sparsity). We provide a (near) matching upper bound for the case $q \in (0, 2)$, which essentially resolves the question: showing that the aforementioned hardness and disappearance of a phase transition apply to all types of sparsity.

**Theorem 2** (Informal; see Theorem 6). *For every $\varepsilon, \gamma$, the sample complexity of robust sparse Gaussian mean testing in the $\varepsilon$-corruption model is*

$$O\left(\frac{m}{\varepsilon^2} \log \frac{ed}{\gamma}\right)$$

*for $q \in (0, 2)$, where $m = \max\{u \in [d] : \gamma^2 u^{\frac{2}{q}-1} \leq s^2\}$.*

Note that, for constant $\gamma$, our upper bound only differs from the lower bound of Theorem 1 by a logarithmic dependence on the effective sparsity $m$. Finally, we note that while the upper bounds from [Li17] and Theorem 2 are achieved by computationally inefficient algorithms (time complexity exponential in $s$), this is actually inherent; indeed, [BB20] recently proved that any computationally efficient algorithm for robust sparse mean *estimation* must have much higher sample complexity, namely $\Omega(s^2)$. A simple inspection of their proof shows that this result extends to the robust *testing* problem.

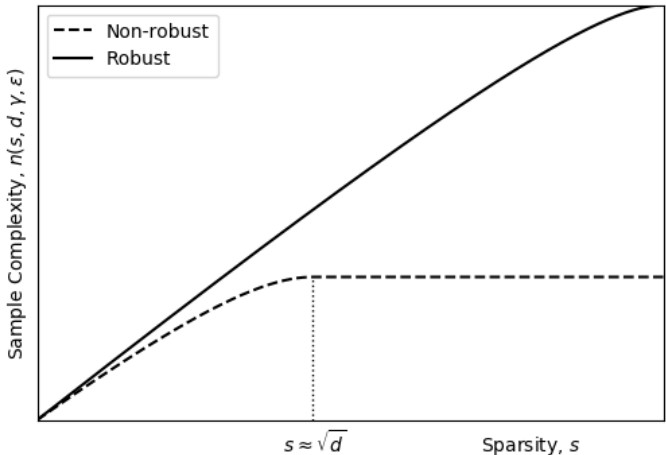

Figure 1: Sample complexity as a function of sparsity for non-robust and robust sparse Gaussian mean testing (the behavior applies to testing in the sparse regression model as well).

**Testing in sparse linear regression model.** From the results in [CCC⁺19] we can deduce that the sample complexity of *non-robust* testing in the sparse linear regression model is given by

$$
n_0(s, d, \gamma) = \begin{cases} \Theta\left(\min\left(\frac{s}{\gamma^2} \log\left(1 + \frac{d}{s^2}\right), \frac{1}{\gamma^4}\right)\right) & \text{if } s < \sqrt{d} \\ \Theta\left(\min\left(\frac{\sqrt{d}}{\gamma^2}, \frac{1}{\gamma^4}\right)\right) & \text{if } s \geq \sqrt{d}. \end{cases}
$$

This expression looks very similar to the sample complexity of sparse Gaussian mean testing, except for an additional $1/\gamma^4$ term. This term is essentially due to the fact that we can ignore the observations $X_i$'s and estimate $\gamma$ just by using $y_i$'s, since their variance is $1 + \gamma^2$. This would require $O(1/\gamma^4)$ samples. Nevertheless, the testing in sparse linear regression still exhibits a phase transition at $s \approx \sqrt{d}$ as was observed in the sparse Gaussian mean testing. It is thus natural to wonder whether the parallels between these two problems extend to the robust setting as well. We show that this is indeed the case: the sample complexity of testing in sparse linear regression significantly increases when introducing the robustness condition.

**Theorem 3** (Informal; see Theorem 7). *For any sufficiently small $\gamma > 0$, $\varepsilon = \frac{\gamma}{C}$ for a sufficiently large $C$, and $s = d^{1-\delta}$ for any $\delta \in (0, 1)$, the sample complexity of testing in sparse linear regression under $\varepsilon$-corruption model is*

$$
\Omega\left(\min\left(s \log d, \frac{1}{\gamma^4}\right)\right).
$$

The tightness of this bound follows from the results in [LSLC20, Theorem 2.1] which states that any algorithm for robust sparse mean estimation can be used for robust sparse linear regression with a polylog$(1/\gamma)$ increase in the sample complexity. Hence the agnostic hypothesis selection via tournaments algorithm in [Li17], which is a statistically optimal algorithm for robust sparse mean estimation works in this case as well.

## 1.2 Our techniques

To establish the lower bounds in Theorem 1, we combine a range of tools from information theory and the literature on robust estimation. First, we argue that it is enough to consider the standard sparsity case ($q = 0$), as this will imply the analogous result for $q \in (0, 2)$. Indeed, the definition of effective sparsity will enable us to deduce that any $x \in \mathbb{R}^d$ with $\|x\|_0 = m$ and $\|x\|_2 = \gamma$ belong to the set $\mathcal{B}_{s,q}$, letting us establish the lower bound given in Theorem 1 for general $q$ from the $q = 0$ case. In this discussion, we therefore focus on standard sparsity, and assume that our observations are from a distribution for which the unknown parameter $\theta$ is in the set $\mathcal{B}_s$.

To simplify further, we note that in order to establish the desired lower bound it suffices to consider the weaker $\varepsilon$-Huber contamination model instead of the (more general) $\varepsilon$-corruption model [DK19]. In

the the $\varepsilon$-Huber contamination model, $n$ i.i.d. samples from a distribution $\mathbf{p}$ are, after contamination, modeled as a set of $n$ i.i.d. samples from a distribution $(1 - \varepsilon)\mathbf{p} + \varepsilon N$, where $N$ is an arbitrary and unknown probability distribution (that is, the adversary is "oblivious:" limited to choosing, ahead of time, a "bad" mixture component $N$ to fool the algorithm). We thus can restrict ourselves, for our lower bound, to the setting where the $n$ i.i.d. samples are from some distribution in $\{(1 - \varepsilon)\mathcal{N}(\theta, I_d) + \varepsilon N_{\varepsilon, \theta} : \|\theta\|_0 \leq s\}$, where we get to design the distributions $N_{\varepsilon, \theta}$. Let $\tilde{\mathcal{C}}_s := \{(1 - \varepsilon)\mathcal{N}(\theta, I_d) + \varepsilon N_{\varepsilon, \theta} : \|\theta\|_0 \leq s, \|\theta\|_2 \geq \gamma\}$ denote the set of $\varepsilon$-Huber contaminated versions of Gaussian distributions whose mean has $\ell_2$ norm greater than $\gamma$. Our goal can be rephrased as choosing as suitable set of parameters $\Theta$ and a corresponding ensemble of distributions $\{\mathbf{q}_\theta\}_{\theta \in \Theta} \subseteq \tilde{\mathcal{C}}_s$, and argue that no algorithm can distinguish $n$ samples drawn from a (randomly chosen) element $\mathbf{q}_\theta$ from $n$ samples drawn from $\mathcal{N}(0, I_d)$.

To do so, define the mixture $\mathbf{q} := \frac{1}{|\Theta|} \sum_\theta \mathbf{q}_\theta^n$. (equivalently $\mathbf{q} = \mathbb{E}_{\theta \sim \mathbf{u}_\Theta}[\mathbf{q}_\theta^n]$). Le Cam's two-point method allows us to reduce the above indistinguishability problem to showing that $\mathrm{d}_{\mathrm{TV}}(\mathbf{q}, \mathcal{N}(0, I_d)^n) = o(1)$, for which, in view of the standard inequality, $\mathrm{d}_{\mathrm{TV}}(\mathbf{q}, \mathbf{p})^2 \leq \frac{1}{4}\chi^2(\mathbf{q} \| \mathbf{p})$, it suffices to show that $\chi^2(\mathbf{q} \| \mathcal{N}(0, I_d)^n) = o(1)$. By the Ingster–Suslina method, this $\chi^2$-divergence between a point and a mixture distribution can then be simplified as

$$1 + \chi^2(\mathbf{q} \| \mathcal{N}(0, I_d)^n) = 1 + \chi_{\mathcal{N}(0, I_d)^n}(\mathbb{E}_\theta[\mathbf{q}_\theta^n], \mathbb{E}_{\theta'}[\mathbf{q}_{\theta'}^n]) = \mathbb{E}_{\theta, \theta'}\left[\left(1 + \chi_{\mathcal{N}(0, I_d)}(\mathbf{q}_\theta, \mathbf{q}_{\theta'})\right)^n\right], \tag{3}$$

where $\theta, \theta' \sim \mathbf{u}_\Theta$, and $\chi_\mathbf{p}(\mathbf{q}, \mathbf{q}') := \int \frac{\mathrm{d}\mathbf{q}\mathrm{d}\mathbf{q}'}{\mathrm{d}\mathbf{p}} - 1$ is the "$\chi^2$-correlation" between $\mathbf{q}$ and $\mathbf{q}'$ with respect to $\mathbf{p}$. Thus, the challenge is to design an ensemble $\{\mathbf{q}_\theta\}$ that yields a good upper bound for the r.h.s. of (3), while still being simple enough to analyze.

Building upon previous works [DKS17, BB20], we define our ensemble $\{\mathbf{q}_\theta\}$ as follows: let

$$\Theta := \left\{\theta \in \left\{-\frac{1}{\sqrt{s}}, 0, \frac{1}{\sqrt{s}}\right\}^d : \|\theta\|_0 = s\right\} \tag{4}$$

and, for $\theta \in \Theta$, $\mathbf{q}_\theta := (1 - \varepsilon)\mathcal{N}(\gamma\theta, I_d) + \varepsilon\mathcal{N}(\mu_\theta, I_d)$, where $\mu_\theta$ is chosen such that $(1 - \varepsilon)\gamma\theta + \varepsilon\mu_\theta = 0$. Note that indeed, for all $\theta \in \Theta$, $\mathbf{q}_\theta \in \tilde{\mathcal{C}}_s$ and $\|\theta\|_2 = 1$. This choice of $\mathbf{q}_\theta$ combined with the above outline will enable us to prove Theorem 1, by carefully upper bounding $\mathbb{E}_{\theta, \theta'}\left[\left(1 + \chi_{\mathcal{N}(0, I_d)}(\mathbf{q}_\theta, \mathbf{q}_{\theta'})\right)^n\right]$ as a function of $n, d$, and $s$.

One can attempt to prove Theorem 3 (the lower bound for testing in the sparse linear regression model) in a similar vein. By restricting ourselves to the $\varepsilon$-Huber contamination model (which only strengthens the resulting lower bound by constraining the adversary), we get the following set of distributions in the sparse linear regression problem: $\{(1 - \varepsilon)\mathcal{N}(0, \Sigma_\theta) + \varepsilon N_{\varepsilon, \theta} : \|\theta\|_0 \leq s\}$, where $\Sigma_\theta := \begin{bmatrix} I_d & \theta \\ \theta^T & 1 + \|\theta\|^2 \end{bmatrix}$. Further defining $\tilde{\mathcal{D}}_s := \{(1 - \varepsilon)\mathcal{N}(0, \Sigma_\theta) + \varepsilon N_{\varepsilon, \theta} : \|\theta\|_0 \leq s, \|\theta\|_2 \geq \gamma\}$, as earlier our problem boils down to finding an ensemble $\{\mathbf{q}_\theta\} \subseteq \tilde{\mathcal{D}}_s$ such that $\chi^2(\mathbf{q} \| \mathcal{N}(0, I_{d+1})^n) = o(1)$ for $\mathbf{q} := \frac{1}{|\Theta|} \sum_\theta \mathbf{q}_\theta^n$.

Notice that in the sparse linear regression model, $y_i$ has marginal distribution $\mathcal{N}(0, 1 + \|\theta\|^2)$ and conditioned on $y_i$ the distribution of $X_i$ is given by $\mathcal{N}\left(\frac{y_i\theta}{1 + \|\theta\|^2}, I - \frac{\theta\theta^T}{1 + \|\theta\|^2}\right)$. This observation, along with the indistinguishability result established while proving Theorem 1 encourage us to choose $\Theta$ as in (4) and $\mathbf{q}_\theta$ in the following manner:

$$\mathbf{q}_\theta(y_i) = \mathcal{N}\left(0, 1 + \gamma^2\right)$$

$$\mathbf{q}_\theta(X_i \mid y_i) = (1 - \varepsilon)\mathcal{N}\left(\frac{y_i\gamma\theta}{1 + \gamma^2}, I - \frac{\theta\theta^T}{1 + \gamma^2}\right) + \varepsilon\mathcal{N}\left(\frac{y_i\mu_\theta}{1 + \gamma^2}, I - \frac{\theta\theta^T}{1 + \gamma^2}\right),$$

where $\mu_\theta$ is chosen such that $(1 - \varepsilon)\gamma\theta + \varepsilon\mu_\theta = 0$. Again, note that $\mathbf{q}_\theta \in \tilde{\mathcal{D}}$. Although this setup looks promising in giving us the right lower bound, it turns out that certain tail events can cause $\chi^2(\mathbf{q} \| \mathcal{N}(0, I_{d+1})^n)$ to blow up to infinity. To circumvent this, one of the remedies available in the literature is to use the *conditional second moment method* [RXZ19, WX18]. In this method, one carefully conditions out the rare events that preclude us from evaluating the $\chi^2$-divergence. Specifically, we define an event $\mathcal{E}$ generated by the random variables $(y_1, y_2, \cdots, y_n)$ such that

$\mathbf{q}(\mathcal{E}^C) = o(1)$. We then define $\mathbf{q}^{\mathcal{E}}$ as the distribution $\mathbf{q}$ conditioned on the "good" event $\mathcal{E}$. That is,

$$\mathbf{q}^{\mathcal{E}}(X, Y) = \mathbf{q}(X, Y \mid \mathcal{E}) = \frac{\mathbb{E}_\theta[\mathbf{q}_\theta(X, Y)\mathbb{1}_{\mathcal{E}}\{Y\}]}{\mathbf{q}(\mathcal{E})}. \tag{5}$$

Note that we have the relation $\mathbf{q} = \mathbf{q}(\mathcal{E})\mathbf{q}^{\mathcal{E}} + \mathbf{q}(\mathcal{E}^C)\mathbf{q}^{\mathcal{E}^C}$. The convexity of total variation distance consequently gives

$$\begin{aligned}
\mathrm{d}_{\mathrm{TV}}(\mathbf{q}, \mathcal{N}(0, I_{d+1})^n) &\leq \mathbf{q}(\mathcal{E})\,\mathrm{d}_{\mathrm{TV}}(\mathbf{q}^{\mathcal{E}}, \mathcal{N}(0, I_{d+1})^n) + \mathbf{q}(\mathcal{E}^C)\,\mathrm{d}_{\mathrm{TV}}(\mathbf{q}^{\mathcal{E}^C}, \mathcal{N}(0, I_{d+1})^n) \\
&\leq \mathrm{d}_{\mathrm{TV}}(\mathbf{q}^{\mathcal{E}}, \mathcal{N}(0, I_{d+1})^n) + \mathbf{q}(\mathcal{E}^C)
\end{aligned}$$

Thus, in view of the fact that $\mathbf{q}(\mathcal{E}^C) = o(1)$, deriving a lower bound reduces to showing that $\mathrm{d}_{\mathrm{TV}}(\mathbf{q}^{\mathcal{E}}, \mathcal{N}(0, I_{d+1})^n)$ is small, for which we saw earlier that it was sufficient (and more convenient) to show that $\chi^2(\mathbf{q}^{\mathcal{E}} \,||\, \mathcal{N}(0, I_{d+1})^n) = o(1)$. This is the roadmap we will follow – first, defining a suitable event $\mathcal{E}$, before showing that $\chi^2(\mathbf{q}^{\mathcal{E}} \,||\, \mathcal{N}(0, I_{d+1})^n) = o(1)$.

The above outlines our approach to proving Theorems 1 and 3. To establish the upper bound in Theorem 2, we show that the "Agnostic hypothesis selection via Tournaments" algorithm [DKK$^+$16] achieves the stated sample complexity. The argument, in turn, is very similar to the proof of the upper bound for robust sparse mean estimation given in [Li17].

## 1.3   Related work

The study of high-dimensional signals with a sparse underlying structure has enjoyed a significant amount of attention in statistics and signal processing for the past few decades. This line of research has led to the discovery of surprising phenomena such as phase transitions and computational hardness in problems involving sparse signals. The main motivation to study sparse signals is that the sample complexity of statistical tasks involving sparsity is typically significantly smaller than that of dense signals, leading to much more data-efficient algorithms. This yields significant savings whenever the problem is expected to exhibit such a sparse structure, e.g., for physical or biological reasons, or due to a specific design choice when engineering a system. Yet, practical scenarios seldom involve noise-free or perfect signals, which effectively destroys the sparsity of the signal one would have capitalized on. This led to the study of these questions under various noise models, in order to understand if one could still see the same type of sample size savings in these settings.

There is a large body of work on the estimation and detection of signals with a sparse structure under noise (e.g., [DJ04, Bar02, Ver12]). Most recently, [CCT17] gave the tight characterization of minimax rates of estimating linear and quadratic functionals of sparse signals under Gaussian noise. The authors also derived the minimum detection level required for reliably testing the $\ell_2$ norm of sparse signals under Gaussian noise. Another prominent sparse signal model studied in the literature is that of sparse linear regression [ITV10, Ver12, RXZ19]. In the non-asymptotic setting, [CCC$^+$19] established the tight sample complexity of testing in the sparse linear regression model. However, these line of works focused on *random* noise, and not the more challenging types of noise allowing for *adversarial* corruptions – what is commonly known as seeking *robust* algorithms.

The systematic study of the robustness of statistical procedures was initiated in the foundational works of Huber [Hub64] and Tukey [Tuk60]. Several statistically optimal procedures were found to break down even under slight model misspecification or sample contamination. Although there was substantial progress in the field of robust statistics, surprisingly, *computationally efficient* procedures remained elusive until recently, even for simple tasks such as high-dimensional mean estimation.

In this regard, the past few years witnessed an incredible progress in algorithmic robust statistics. A line of work initiated by [DKK$^+$16] and [LRV16] provided computationally efficient optimal robust estimators for various estimation tasks in high dimension [DKK$^+$19]. The surprising upshot from these papers is that even with robustness requirements, the sample complexity of many high-dimensional estimation tasks remains essentially the same, at no extra computational cost: i.e., that robustness and computational efficiency are not at odds for those estimation tasks. And still, a subsequent result of [DKS17] shows that, surprisingly, imposing robustness constraints *does* significantly increase the sample complexity of the Gaussian mean *testing* problem, and makes the sample complexity as large as that of Gaussian mean estimation: that is, for testing, robustness comes at a very high cost, and negates the usual savings that testing allows over estimation. Extending their

results on mean testing, [DK21] shows the analogue in the case of Gaussian covariance testing under the Frobenius norm. Yet, those striking results focus on testing *dense* parameters; our work seeks to combine the two lines of work – inference under sparsity guarantees, and robustness – to understand if an analogous jump in sample complexity occurs for testing in high-dimensional *sparse* models.

We further note that several works have shown evidence for the existence of statistical-computation gaps in estimation problems with sparse signal structure [BR13, CW20]. [DKS17] gave the first evidence for the presence of an $s$ to $s^2$ statistical-computation gap in robust sparse mean estimation, in the form of a Statistical Query (SQ) lower bound (i.e., for a restricted type of algorithms). This was complemented by [Li17] and [BDLS17], which gave computationally efficient algorithms for robust sparse estimation achieving $O(s^2)$ sample complexity. Finally, [BB20] recently used average-case reductions to prove the algorithmic hardness of robust sparse mean estimation and robust sparse linear regression.

## 1.4 Preliminaries and notation

Given a probability distribution $\mathbf{p}$ and integer $n \geq 1$, we denote by $\mathbf{p}^n$ the $n$-fold product distribution with marginals $\mathbf{p}$, and given a set of distributions $\mathcal{D}$ write $\mathcal{D}^n = \{ \mathbf{p}^n : \mathbf{p} \in \mathcal{D} \}$. For $0 < q \leq 2$ and $r > 0$, we let $\mathcal{B}_{q,r} = \{\theta \in \mathbb{R}^d : \|\theta\|_q \leq r\}$ the $\ell_q$ ball of radius $r$. We say a vector $\theta \in \mathbb{R}^d$ is *s-sparse* if $\|\theta\|_0 \leq s$; for $q \in (0, 2]$, we will accordingly say that $\theta$ is *s-sparse in $\ell_q$ norm* whenever $\|\theta\|_q \leq s$ (note that $s$ need not be an integer). We denote the uniform distribution over a set $S$ by $\mathbf{u}_S$. The delta measure (point mass) at an element $x$ is denoted by $\delta_x$. The total variation distance between two distributions $\nu$ and $\mu$ is defined as: $\mathrm{d}_{\mathrm{TV}}(\nu, \mu) = \frac{1}{2} \int |d\nu - d\mu|$, which is to be interpreted as $\frac{1}{2} \int \left| \frac{d\nu}{d\lambda} - \frac{d\mu}{d\lambda} \right| d\lambda$, where $\lambda$ is any distribution dominating both $\nu$ and $\mu$; equivalently, $\mathrm{d}_{\mathrm{TV}}(\nu, \mu) = \sup_S(\nu(S) - \mu(S))$ where the supremum is over all measurable sets $S$. The $\chi^2$-*divergence between $\nu$ and $\mu$* is given by $\chi^2(\nu \,\|\, \mu) = \int \frac{d\nu d\nu}{d\mu} - 1$, and satisfies $\mathrm{d}_{\mathrm{TV}}(\nu, \mu) \leq \frac{1}{2}\sqrt{\chi^2(\nu \,\|\, \mu)}$. Finally, given a reference probability measure $\mu$, the $\chi^2$-*correlation between $\nu$ and $\lambda$ with respect to* $\mu$ is defined as $\chi_\mu(\nu, \lambda) = \int \frac{d\nu d\lambda}{d\mu} - 1$; note that $\chi^2(\nu \,\|\, \mu) = \chi_\mu(\nu, \nu)$.

We will also rely in several occasions on the following technical lemma due to Cai, Ma, and Wu, which helps us bound the moment generating function of the square of a random sum of Rademacher random variables, which will arise when we consider sparse priors in our lower bounds.

**Lemma 1** (Lemma 1 in [CMW15]). *Fix $d \in \mathbb{N}$ and $s \in [d]$. Let $H \sim \mathrm{Hypergeometric}(d, s, s)$, and let $\xi_1, \xi_2, \cdots, \xi_s$ be i.i.d. Rademacher. Define $Y := \sum_{i=1}^H \xi_i$. Then there exists a function $\tau \colon (0, \frac{1}{36}) \to (1, \infty)$ with $\tau(0^+)=1$, such that for any $0 < b < \frac{1}{36}$, and $\lambda := \frac{b}{s} \log \frac{ed}{s}$,*

$$\mathbb{E}\left[\exp\left(\lambda Y^2\right)\right] \leq \tau(b).$$

## 2  Main results

In this section, we formally state and give proofs for the theorems outlined informally in Section 1.1. Recall that a lower bound or the hardness of hypothesis testing between two distributions $\mathbf{p}$ and $\mathbf{q}$ can be characterized by the total variation distance between them. Indeed, by the Pearson–Neyman lemma, if there exists test which successfully distinguishes between two distributions $\mathbf{p}_n$ and $\mathbf{q}_n$ (which in our case will correspond to distributions over $n$ tuples of i.i.d. samples) with probability at least $2/3$, then one must have $\mathrm{d}_{\mathrm{TV}}(\mathbf{p}_n, \mathbf{q}_n) \geq 1/3$. Hence, to prove indistinguishability for a given $n$, it suffices to show $\mathrm{d}_{\mathrm{TV}}(\mathbf{p}_n, \mathbf{q}_n) = o(1)$; since $\mathrm{d}_{\mathrm{TV}}(\mathbf{p}_n, \mathbf{q}_n)^2 \leq \frac{1}{4}\chi^2(\mathbf{q}_n \,\|\, \mathbf{p}_n)$, one can then focus on showing $\chi^2(\mathbf{q}_n \,\|\, \mathbf{p}_n) = o(1)$.

**Sparse Gaussian mean testing.**  We now state and prove our results related to sparse Gaussian mean testing. First, we derive the lower bounds in Theorem 1 using the techniques outlined in Section 1.2 and then show a matching upper bound (up to logarithmic factors) for $q > 0$ case.

**Theorem 4.** *Let $\varepsilon, \gamma > 0$ be fixed. Let $X_1, X_2, \cdots, X_n$ be i.i.d. samples from an unknown distribution $\mathbf{p}$. Moreover, suppose an $\varepsilon$-fraction of these $n$ samples are arbitrarily corrupted. Then, if there exists an algorithm that distinguishes between the cases $\mathbf{p} = \mathcal{N}(0, I_d)$ and $\mathbf{p} \in \{\mathcal{N}(\theta, I_d) : \|\theta\|_2 \geq \gamma, \|\theta\|_0 \leq s\}$ with probability greater than $2/3$, we must have $n = \Omega\left(s \log \frac{ed}{s}\right)$.*

To prove this theorem, we will require the following lemma due to Diakonikolas, Kane, and Stewart [DKS17].

**Lemma 2** ([DKS17, Lemma 6.9])**.** *Fix $\theta, \theta' \in \mathbb{R}^d$, $\varepsilon \in (0, 1/3]$, and let $\mathbf{q}_\theta$ be defined as*

$$\mathbf{q}_\theta := (1 - \varepsilon)\mathcal{N}(\theta, I_d) + \varepsilon\mathcal{N}\left(-\frac{(1 - \varepsilon)}{\varepsilon}\theta, I_d\right).$$

*Then,*

$$1 + |\chi_{\mathcal{N}(0, I_d)}(\mathbf{q}_\theta, \mathbf{q}_{\theta'})| \leq \exp\left(\frac{\langle\theta, \theta'\rangle^2}{\varepsilon^4}\right).$$

With this in hand, we are now able to establish Theorem 4.

*Proof of Theorem 4.* For the purpose of deriving a lower bound, it is enough to consider the weaker $\varepsilon$-Huber model for the corruption of samples, as then a lower bound on the sample complexity of the hypothesis testing problem in this setting will also be a lower bound for robust sparse Gaussian mean testing in the adversarial one.

$$\begin{aligned} \mathcal{H}_0 &: X_i \sim \mathcal{N}(0, I_d) \\ \mathcal{H}_1 &: X_i \sim (1 - \varepsilon)\mathcal{N}(\theta, I_d) + \varepsilon N_{\varepsilon,\theta} \quad \text{s. t.} \quad \|\theta\|_2 \geq \gamma \text{ and } \|\theta\|_0 \leq s, \end{aligned}$$

for $1 \leq i \leq n$. Here the distributions $N_{\varepsilon,\theta}$ need to be chosen appropriately.

Let $B = \{\beta \in \{-1, 0, 1\}^d : \|\beta\|_0 = s\}$ and $\beta \sim \mathbf{u}_B$. We can think of $\beta$ as being generated according to the following process: First pick an element uniformly from the set $\{b \in \{0, 1\}^d : \|b\|_0 = s\}$ and then set the non-zero elements to be i.i.d. Rademacher random variables.

Define $\Theta := \frac{1}{\sqrt{s}}B$ and $\theta := \frac{1}{\sqrt{s}}\beta$, so that $\|\theta\|_2 = 1$. Define $\mathbf{q}_\theta$ as

$$\mathbf{q}_\theta = (1 - \varepsilon)\mathcal{N}(\gamma\theta, I_d) + \varepsilon\mathcal{N}\left(-\frac{(1 - \varepsilon)}{\varepsilon}\gamma\theta, I_d\right).$$

It is immediate to see that $\mathbf{q}_\theta \in \mathcal{H}_1$ for all $\theta \in \Theta$. Let $\mathbf{q} := \mathbb{E}_{\theta \sim \mathbf{u}_\Theta}[\mathbf{q}_\theta^n]$. Then by (3),

$$1 + \chi^2(\mathbf{q} \,\|\, \mathcal{N}(0, I_d)^n) = \mathbb{E}_{\theta,\theta'}\left[\left(1 + \chi_{\mathcal{N}(0, I_d)}(\mathbf{q}_\theta, \mathbf{q}_{\theta'})\right)^n\right].$$

By Lemma 2, we get

$$1 + \chi_{\mathcal{N}(0, I_d)}(\mathbf{q}_\theta, \mathbf{q}_{\theta'}) \leq \exp\left(\frac{\gamma^4\langle\theta, \theta'\rangle^2}{\varepsilon^4}\right) = \exp\left(\frac{\gamma^4\langle\beta, \beta'\rangle^2}{\varepsilon^4 s^2}\right).$$

Hence,

$$1 + \chi^2(\mathbf{q} \,\|\, \mathcal{N}(0, I_d)^n) \leq \mathbb{E}_{\beta,\beta'}\left[\exp\left(\frac{n\gamma^4\langle\beta, \beta'\rangle^2}{\varepsilon^4 s^2}\right)\right]. \tag{6}$$

Now by symmetry of the problem, it suffices to evaluate the expectation for any fixed $\beta'$, say to $\beta_0 := (\underbrace{1, 1, \cdots, 1}_{s}, 0, 0, \cdots, 0)$. We can characterize $\langle\beta, \beta_0\rangle$ as follows: let $H$ be a random variable with distribution Hypergeometric$(d, s, s)$ and let $\xi_1, \xi_2, \cdots, \xi_s$ be i.i.d. Rademacher random variables independent of $H$. Then, we have $Y := \langle\beta, \beta_0\rangle = \sum_{i=1}^{H} \xi_i$, where $Y$ can be thought of as a symmetric random walk with Hypergeometric stopping time. We can then rewrite (6) as

$$1 + \chi^2(\mathbf{q} \,\|\, \mathcal{N}(0, I_d)^n) \leq \mathbb{E}\left[\exp\left(\frac{n\gamma^4 Y^2}{\varepsilon^4 s^2}\right)\right].$$

By Lemma 1, if $n \leq c\frac{\varepsilon^4}{\gamma^4} \cdot s \log \frac{ed}{s}$ for a sufficiently small $c > 0$ (e.g., $c < 1/36$ suffices),

$$\chi^2(\mathbf{q} \,\|\, \mathcal{N}(0, I_d)^n) \leq \mathbb{E}\left[\exp\left(\frac{n\gamma^4 Y^2}{\varepsilon^4 s^2}\right)\right] - 1 \leq \tau(c) - 1.$$

Since $\tau$ is continuous at $0^+$, we can choose $c > 0$ to make $\tau(c) - 1$ arbitrarily small. This implies that it is impossible to distinguish between $\mathcal{H}_0$ and $\mathcal{H}_1$ with high probability if $n = o\left(s \log \frac{ed}{s}\right)$, establishing the theorem. $\qquad\square$

Next, we extend this lower bound to other sparsity notions, namely, with respect to the $\ell_q$-norms.

**Theorem 5.** *Let $\varepsilon, \gamma > 0$ be fixed, and $q \in (0, 2)$. Let $X_1, X_2, \cdots, X_n$ be i.i.d. samples from an unknown distribution $\mathbf{p}$. Moreover, suppose an $\varepsilon$-fraction of these $n$ samples are arbitrarily corrupted. Then, if there exists an algorithm that distinguishes between the cases $\mathbf{p} = \mathcal{N}(0, I_d)$ and $\mathbf{p} \in \{\mathcal{N}(\theta, I_d) : \|\theta\|_2 \geq \gamma, \|\theta\|_q \leq s\}$ with probability greater than 2/3, we must have $n = \Omega\left(m \log \frac{ed}{m}\right)$, where $m$ is the effective sparsity defined as: $m := \max\{u \in [d] : \gamma^2 u^{\frac{2}{q}-1} \leq s^2\}$.*

*Proof.* The parameter set of the alternative hypothesis in this problem is $\Theta = \{\theta \in \mathbb{R}^d : \|\theta\|_q \leq s \text{ and } \|\theta\|_2 \geq \gamma\}$. To conclude, it suffices to show that every $m$-sparse (in $\ell_0$ sense) vector with $\ell_2$ norm equal to $\gamma$ belongs to $\Theta$, and then appeal to the proof of Theorem 4 (whose hard instances had mean with magnitude exactly $\gamma$). To do so, let $x \in \mathbb{R}^d$ be $m$-sparse such that $\|x\|_2 = \gamma$. Then,

$$\|x\|_q^2 = \left(\sum_{i=1}^d |x_i|^q\right)^{\frac{2}{q}} \stackrel{\text{(convexity)}}{\leq} m^{\frac{2}{q}-1}\|x\|_2^2 = m^{\frac{2}{q}-1}\gamma^2 \stackrel{\text{(a)}}{\leq} s^2,$$

where (a) is due to the definition of $m$, and the convexity step follows from $\frac{q}{2} \leq 1$. Hence $x \in \Theta$. Thus the lower bound in Theorem 4 holds here with $m$ replacing $s$. $\qquad\square$

We will now proceed to show that the lower bounds in Theorems 4 and 5 are tight up to a logarithmic factor. First, we note that the lower bound given in Theorem 4 is optimal due to the result that the sample complexity of robust sparse Gaussian mean estimation is upper bounded by the same value [DK19, Li17], and the folklore fact that testing is no harder than estimation. So we need to prove the upper bound only for the cases where $q > 0$. We will show that the sample complexity lower bound given in Theorem 5 is tight by proving that the algorithm for robust sparse Gaussian mean estimation given in [Li17] works in the $\ell_q$-norm constrained case as well (and so, again, the upper bound for testing will follow from the upper bound for estimation), in the regime $\gamma = \Theta(\varepsilon)$. Intuitively, this is because all but $m$ coordinates of a vector $\theta$ would be small if $\|\theta\|_q \leq s$ and $\|\theta\|_2 = \gamma$. Our proof is similar to the proof of [Li17, Fact A.1] with a minor modification. We defer its proof to the supplemental.

**Theorem 6.** *Let $\varepsilon, \delta > 0$ and, $q \in (0, 2)$ be fixed. Let $X_1, X_2, \cdots, X_n$ be i.i.d. samples from an unknown distribution $\mathcal{N}(\mu, I_d)$, where $\|\mu\|_q \leq s$. Moreover, suppose an $\varepsilon$-fraction of these $n$ samples are arbitrarily corrupted. Then, there exists an algorithm that, for $n = O\left(\frac{1}{\varepsilon^2}\left(m \log \frac{d}{\varepsilon} + \log \frac{1}{\delta}\right)\right)$, upon being given the $X_1, \ldots, X_n$ outputs a $\mu'$ such that $\|\mu' - \mu\|_2 \leq O(\varepsilon)$ with probability $1 - \delta$. Here $m$ is the effective sparsity defined as: $m = \max\{u \in [d] : \varepsilon^2 u^{\frac{2}{q}-1} \leq s^2\}$.*

**Testing in sparse linear regression model.** Lastly, we state the formal version of Theorem 3, and provide an outline of its proof (see supplementary for a complete proof).

**Theorem 7.** *Let $\gamma > 0$ be sufficiently small, $\varepsilon = \frac{\gamma}{C}$ for a sufficiently large $C$, and $s = d^{1-\delta}$ for some $\delta \in (0, 1)$. Let $(X_1, y_1), (X_2, y_2), \cdots, (X_n, y_n)$ be i.i.d. samples obtained from the sparse linear regression model described in (2). Moreover, suppose an $\varepsilon$-fraction of these $n$ samples are arbitrarily corrupted. Then, if there exists an algorithm that distinguishes between the cases $\|\theta\|_2 = 0$ and $\|\theta\|_2 \geq \gamma$ with probability greater than 2/3, we must have $n = \Omega\left(\min\left(s \log d, \frac{1}{\gamma^4}\right)\right)$.*

*Proof outline.* Let $X$ denote $(X_1, X_2, \cdots, X_n)$ and $Y$ denote $(y_1, y_2, \cdots, y_n)$. Let $\Theta = \{\theta \in \{-\frac{1}{\sqrt{s}}, 0, \frac{1}{\sqrt{s}}\}^d : \|\theta\|_0 = s\}$ and $p = \frac{1-\varepsilon}{\varepsilon}$. We define hypotheses $\mathcal{H}_0$ and $\mathcal{H}_1$ as follows:

$$\mathcal{H}_0 : \quad (X_i, y_i) \stackrel{\text{i.i.d.}}{\sim} \mathbf{p} = \mathcal{N}(0, I_{d+1})$$

$$\mathcal{H}_1 : \quad \theta \sim \mathbf{q}(\theta) = \mathbf{u}_\Theta; \quad y_i \stackrel{\text{i.i.d.}}{\sim} \mathbf{q}(y_i) = \mathcal{N}(0, 1 + \gamma^2)$$

$$\mathbf{q}(X_i|y_i, \theta) = \mathbf{q}_\theta(X_i|y_i) = \mathbb{E}_b\left[\mathcal{N}\left(\frac{by_i\gamma\theta}{1+\gamma^2}, I - \frac{\gamma^2\theta\theta^T}{1+\gamma^2}\right)\right], \text{ where } b \sim (1-\varepsilon)\delta_1 + \varepsilon\delta_{-p}.$$

Here, we denote the probability distribution under the hypotheses $\mathcal{H}_0$ by $\mathbf{p}$ and $\mathcal{H}_1$ by $\mathbf{q}$. Notice that $\mathbf{q}_\theta$ is an $\varepsilon$-Huber contaminated version of $\mathcal{N}(0, \Sigma_{\gamma\theta})$, where $\Sigma_{\gamma\theta} = \begin{bmatrix} I_d & \gamma\theta \\ \gamma\theta^T & 1+\gamma^2 \end{bmatrix}$ and thus a

lower bound on the sample complexity of this hypothesis testing task would be a lower bound for the robust testing in sparse linear regression model. A natural approach to prove the impossibility of detection between $\mathcal{H}_0$ and $\mathcal{H}_1$ would be to try and show that $\chi^2(\mathbf{q} \,\|\, \mathbf{p}) = o(1)$. However, as previously discussed, this method does not yield a useful lower bound for this problem, due to low-probability events which cause $\chi^2(\mathbf{q} \,\|\, \mathbf{p})$ to blow up. Instead, as alluded to in Section 1.2, we evaluate $\chi^2(\mathbf{q}^{\mathcal{E}} \,\|\, \mathbf{p})$, where we define an event $\mathcal{E}$ as

$$\mathcal{E} = \left\{ Y \in \mathbb{R}^d : \frac{\|Y\|^2}{n} \leq 2 + \frac{\log\log d}{n} \right\}, \tag{7}$$

and the probability distribution $\mathbf{q}^{\mathcal{E}}$ is given by (5). Evaluating $\chi^2(\mathbf{q}^{\mathcal{E}} \,\|\, \mathbf{p})$ gives

$$1 + \chi^2(\mathbf{q}^{\mathcal{E}} \,\|\, \mathbf{p}) = \frac{1}{\mathbf{q}(\mathcal{E})^2} \mathbb{E}_{\theta,\theta'} \left[ \underbrace{\mathbb{E}_{Y \sim \mathbf{p}} \left[ \frac{\mathbf{q}(Y)^2}{\mathbf{p}(Y)^2} \mathbb{E}_{X \sim \mathbf{p}} \left[ \frac{\mathbf{q}_\theta(X|Y)\mathbf{q}_{\theta'}(X|Y)}{\mathbf{p}(X)^2} \right] \mathbb{1}_{\mathcal{E}}\{Y\} \right]}_{:=Z} \right]. \tag{8}$$

We compute $\mathbb{E}_{\theta,\theta'}[Z]$ by splitting it into two cases depending on whether $|\langle\theta,\theta'\rangle| \leq \tau$ or $|\langle\theta,\theta'\rangle| > \tau$, where $\tau = \frac{\varepsilon^2}{4e\gamma^2 \log(d)}$. This leads to the result that $\chi^2(\mathbf{q}^{\mathcal{E}} \,\|\, \mathbf{p}) = o(1)$ if $n = o\left(\min\left(s\log d, \frac{1}{\gamma^4}\right)\right)$. We then show that $\mathcal{E}$ is asymptotically a high-probability event, which allows us to argue that $\mathrm{d}_{\mathrm{TV}}(\mathbf{p}, \mathbf{q}) = o(1)$, thereby proving Theorem 7. $\qquad\square$

## 3   Discussion and Future Work

In this section, we discuss some of the limitations of our results, which we believe are ground for possible future work. Firstly, we suspect that our lower bounds are not tight with respect to the parameters $\gamma$ and $\varepsilon$. The dependence on $\gamma$ and $\varepsilon$ in Theorems 4 and 5, is $\Omega(\varepsilon^4/\gamma^4)$. While the exact dependence on these parameters is still an open problem even for robust Gaussian mean testing (non-sparse), we conjecture that the actual dependence on these parameters should scale as $\varepsilon^2/\gamma^4$, and hence believe that our dependence on these parameters is suboptimal. In the current proof, the bottleneck appears while obtaining an upper bound for the chi-squared correlation between two Huber-contaminated distributions, and we suspect that more advanced techniques might be required to get the right dependence on $\gamma$ and $\varepsilon$.

Furthermore, we restricted ourselves to the case when the covariance of the Gaussian distributions under consideration are identity matrices ("spherical Gaussians"); of course, handling the case of arbitrary covariances is a natural and important question. We believe that deriving the sample complexity in the case of non-identity (and unknown) covariance matrices would require significant additional effort, as well as new techniques and ideas. It is worth pointing out that we are not aware of any work addressing the sample complexity results for non-identity covariances even in the non-robust (but sparse) setting.

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
