# Robust Testing in High-Dimensional Sparse Models
## Supplementary (full version)

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

 lemmas: the first will be useful to bound the $\chi^2$ correlation between Gaussians.

**Lemma 1.** *Let $\phi_{\mu,\Sigma}$ denote the density function of $\mathcal{N}(\mu, \Sigma)$. Then,*

$$\mathbb{E}_{X \sim \phi_{0,I}}\left[ \frac{\phi_{\mu_1,\Sigma_1}(X)\phi_{\mu_2,\Sigma_2}(X)}{\phi_{0,I}^2(X)} \right] = \frac{\exp\left( \frac{1}{2}\left( \mu'^T A^{-1}\mu' - \mu_1^T\Sigma_1\mu_1 - \mu_2^T\Sigma_2\mu_2 \right) \right)}{\det(A)^{\frac{1}{2}}\det(\Sigma_1\Sigma_2)^{\frac{1}{2}}},$$

*where $A := \Sigma_1^{-1} + \Sigma_2^{-1} - I$ and $\mu' := \Sigma_1^{-1}\mu_1 + \Sigma_2^{-1}\mu_2$.*

*Proof.* The l.h.s. can be written explicitly as

$$\frac{1}{(2\pi)^{d/2}\det(\Sigma_1\Sigma_2)^{1/2}} \int_{\mathbb{R}^d} \frac{\exp\left( -\frac{1}{2}\left( (x-\mu_1)^T\Sigma_1^{-1}(x-\mu_1) + (x-\mu_2)^T\Sigma_2^{-1}(x-\mu_2) \right) \right)}{\exp\left( -\frac{1}{2}x^Tx \right)} dx.$$

Integrating this quantity by completing the square gives the result. □

The second, due to Cai, Ma, and Wu, will let us bound the moment generating function of the square of a random sum of Rademacher random variables, which will arise when we consider sparse priors in our lower bounds.

**Lemma 2** (Lemma 1 in [CMW15]). *Fix $d \in \mathbb{N}$ and $s \in [d]$. Let $H \sim \text{Hypergeometric}(d, s, s)$, and let $\xi_1, \xi_2, \cdots, \xi_s$ be i.i.d. Rademacher. Define the random variable $Y$ as*

$$Y := \sum_{i=1}^{H} \xi_i.$$

*Then there exists a function $\tau \colon (0, \frac{1}{36}) \to (1, \infty)$ with $\tau(0^+)=1$, such that for any $0 < b < \frac{1}{36}$,*

$$\mathbb{E}\big[\exp\left(\lambda Y^2\right)\big] \leq \tau(b),$$

*where $\lambda := \frac{b}{s} \log \frac{ed}{s}$.*

Finally, we will require the two facts below on Gaussians and Hypergeometric distributions.

**Lemma 3.** *Let $P$ be the density function of $\mathcal{N}\left(0, 1 + \gamma^2\right)$ for some $0 \leq \gamma \leq 1/\sqrt{3}$, and $Q$ be the density function of $\mathcal{N}(0, 1)$. Then*

$$\mathbb{E}_{y \sim Q}\left[\left(\frac{P(y)}{Q(y)}\right)^2 y^{2\ell}\right] \leq \frac{c \cdot 4^\ell \ell^\ell}{\sqrt{1 - \gamma^4}},$$

*where $c > 0$ is a universal constant.*

*Proof.* We have

$$\left(\frac{P(y)}{Q(y)}\right)^2 = \frac{1}{1 + \gamma^2} \exp\left(\frac{\gamma^2 y^2}{1 + \gamma^2}\right).$$

Therefore,

$$\mathbb{E}_{y \sim Q}\left[\left(\frac{P(y)}{Q(y)}\right)^2 y^{2\ell}\right] = \frac{1}{\sqrt{2\pi}} \frac{1}{1 + \gamma^2} \int_{\mathbb{R}} y^{2\ell} \exp\left(\frac{-y^2}{2}\left(1 - \frac{2\gamma^2}{1 + \gamma^2}\right)\right)$$

$$= \frac{1}{\sqrt{1 - \gamma^4}} \left(\frac{1 + \gamma^2}{1 - \gamma^2}\right)^\ell \mathbb{E}_{y \sim Q}\left[y^{2\ell}\right]$$

$$\overset{(a)}{\leq} \frac{c}{\sqrt{1 - \gamma^4}} 4^\ell \ell^\ell,$$

where in (a) we used the facts that, for all $\ell \geq 0$, $\mathbb{E}_{y \sim \mathcal{N}(0,1)}\big[y^{2\ell}\big] \leq c(2\ell)^\ell$ for some absolute constant $c > 0$; and that $\gamma \in [0, 1/\sqrt{3}]$. $\qquad \square$

**Lemma 4.** *Let $H \sim \text{Hypergeometric}(d, s, s)$. Then,*

$$\Pr[H = h] \leq \left(\frac{es^2}{h(d - s + 1)}\right)^h.$$

*Proof.* This directly follows from bounds on binomial coefficients:

$$\Pr[H = h] = \binom{s}{h} \frac{\binom{d-s}{s-h}}{\binom{d}{s}} \leq \left(\frac{es}{h}\right)^h \left(\frac{s}{d - s + 1}\right)^h = \left(\frac{es^2}{h(d - s + 1)}\right)^h.$$

$\qquad \square$

# 3 Main Results and Proofs

In this section, we formally state and give proofs for the theorems outlined informally in Section 1.1. Recall that a lower bound or the hardness of hypothesis testing between two distributions $\mathbf{p}$ and $\mathbf{q}$ can be characterized by the total variation distance between them. Indeed, by the Pearson-Neyman lemma, if there exists test which successfully distinguishes between two distributions $\mathbf{p}_n$ and $\mathbf{q}_n$ (which in our case will correspond to distributions over $n$ tuples of i.i.d. samples) with probability at least $2/3$, then one must have $d_{\mathrm{TV}}(\mathbf{p}_n, \mathbf{q}_n) \geq 1/3$. Hence, to prove indistinguishability for a given $n$, it suffices to show $d_{\mathrm{TV}}(\mathbf{p}_n, \mathbf{q}_n) = o(1)$; since $d_{\mathrm{TV}}(\mathbf{p}_n, \mathbf{q}_n)^2 \leq \frac{1}{4}\chi^2(\mathbf{q}_n \| \mathbf{p}_n)$, one can then focus on showing $\chi^2(\mathbf{q}_n \| \mathbf{p}_n) = o(1)$.

In our problems, we formulate $\mathbf{p}_n$ as a product distribution (product of high-dimensional Gaussians) and $\mathbf{q}_n$ as a mixture distribution. In such cases, the following tensorization property of $\chi^2$-divergence helps in upper bounding it: Let $\mathbf{q} := \int \mathbf{q}_\theta^n d\theta$ be a mixture distribution. Then,

$$
\begin{aligned}
1 + \chi^2(\mathbf{q} \| \mathbf{p}^n) &= \int_{\mathbb{R}^n} \frac{\mathbf{q}(x)^2}{\mathbf{p}^n(x)} dx = \int_{\mathbb{R}^n} \frac{\int \mathbf{q}_\theta^n(x) d\theta \int \mathbf{q}_{\theta'}^n(x) d\theta'}{\mathbf{p}^n(x)} dx \\
&= \int_\theta \int_{\theta'} \int_{\mathbb{R}^n} \frac{\mathbf{q}_\theta^n(x)\mathbf{q}_{\theta'}^n(x)}{\mathbf{p}^n(x)} dx \, d\theta \, d\theta' = \int_\theta \int_{\theta'} \left( \int_{\mathbb{R}} \frac{\mathbf{q}_\theta(x)\mathbf{q}_{\theta'}(x)}{\mathbf{p}(x)} dx \right)^n d\theta \, d\theta' \\
&= \mathbb{E}_{\theta,\theta'}[(1 + \chi_{\mathbf{p}}(\mathbf{q}_\theta, \mathbf{q}_{\theta'}))^n].
\end{aligned}
$$

This approach is widely known as the Ingster–Suslina method [IS03], and is the starting point of many minimax lower bounds.

## 3.1 Sparse Gaussian Mean Testing

In this section, we state and prove our results related to sparse Gaussian mean testing. First, we derive the lower bounds in Theorem 1 using the techniques outlined in Section 1.2 and then show a matching upper bound (up to logarithmic factors) for $q > 0$ case.

**Theorem 4.** *Let $\varepsilon, \gamma > 0$ be fixed. Let $X_1, X_2, \cdots, X_n$ be i.i.d. samples from an unknown distribution $\mathbf{p}$. Moreover, suppose an $\varepsilon$-fraction of these $n$ samples are arbitrarily corrupted. Then, if there exists an algorithm that distinguishes between the cases $\mathbf{p} = \mathcal{N}(0, I_d)$ and $\mathbf{p} \in \{\mathcal{N}(\theta, I_d) : \|\theta\|_2 \geq \gamma, \|\theta\|_0 \leq s\}$ with probability greater than $2/3$, we must have $n = \Omega\left(s \log \frac{ed}{s}\right)$.*

To prove this theorem, we will require the following lemma due to Diakonikolas, Kane, and Stewart [DKS17]; we provide below an alternative proof, which we believe is simpler than the original.

**Lemma 5** ([DKS17, Lemma 6.9]). *Fix $\theta, \theta' \in \mathbb{R}^d$, $\varepsilon \in (0, 1/3]$, and let $\mathbf{q}_\theta$ be defined as*

$$
\mathbf{q}_\theta := (1 - \varepsilon)\mathcal{N}(\theta, I_d) + \varepsilon\mathcal{N}\left(-\frac{(1 - \varepsilon)}{\varepsilon}\theta, I_d\right).
$$

*Then,*

$$
1 + |\chi_{\mathcal{N}(0, I_d)}(\mathbf{q}_\theta, \mathbf{q}_{\theta'})| \leq \exp\left(\frac{\langle \theta, \theta' \rangle^2}{\varepsilon^4}\right).
$$

*Proof.* We here provide a simple proof. Let $p := \frac{1-\varepsilon}{\varepsilon}$, which is at least $2$ since $\varepsilon \leq 1/3$. The distribution $\mathbf{q}_\theta$ can be written as follows:

$$
\mathbf{q}_\theta = \mathbb{E}_b[\mathcal{N}(b\theta, I_d)],
$$

where $b \sim (1 - \varepsilon)\delta_1 + \varepsilon\delta_{-p}$. Let $\phi_{\mu,\Sigma}$ denote the density function of $\mathcal{N}(\mu, \Sigma)$. Then,

$$
\begin{aligned}
\chi_{\mathcal{N}(0, I_d)}(\mathbf{q}_\theta, \mathbf{q}_{\theta'}) &= \mathbb{E}_{X \sim \mathcal{N}(0, I_d)}\left[ \frac{\mathbb{E}_b[\phi_{b\theta, I_d}(X)]\mathbb{E}_{b'}[\phi_{b'\theta', I_d}(X)]}{\phi_{0, I_d}^2(X)} \right] - 1 \\
&= \mathbb{E}_{bb'}\left[ \mathbb{E}_{X \sim \mathcal{N}(0, I_d)}\left[ \frac{\phi_{b\theta, I_d}(X)\phi_{b'\theta', I_d}(X)}{\phi_{0, I_d}^2(X)} \right] \right] - 1.
\end{aligned}
$$

By Lemma 1, we have

$$\mathbb{E}_{X \sim \mathcal{N}(0, I_d)} \left[ \frac{\phi_{b\theta, I_d}(X) \phi_{b'\theta', I_d}(X)}{\phi_{0, I_d}^2(X)} \right] = \exp\left( bb' \langle \theta, \theta' \rangle \right),$$

from which

$$\chi_{\mathcal{N}(0, I_d)}(\mathbf{q}_\theta, \mathbf{q}_{\theta'}) = \mathbb{E}_{bb'} \left[ \exp\left( bb' \langle \theta, \theta' \rangle \right) \right] - 1 = \sum_{\ell=1}^{\infty} \frac{\mathbb{E}_b \left[ b^\ell \right]^2 \langle \theta, \theta' \rangle^\ell}{\ell!}.$$

Note that $\mathbb{E}[b] = 0$, and that $\left| \mathbb{E}_b \left[ b^\ell \right] \right| = \left| (1 - \varepsilon) + (-1)^\ell p^\ell \varepsilon \right| = (1 - \varepsilon) \left| 1 + (-1)^\ell p^{\ell-1} \right| \le 2p^{\ell-1}$. Thus, we have

$$
\begin{aligned}
\left| \chi_{\mathcal{N}(0, I_d)}(\mathbf{q}_\theta, \mathbf{q}_{\theta'}) \right| &\le \frac{4}{p^2} \sum_{\ell=2}^{\infty} \frac{p^{2\ell} |\langle \theta, \theta' \rangle|^\ell}{\ell!} \\
&= \frac{4}{p^2} \left( \exp\left( p^2 |\langle \theta, \theta' \rangle| \right) - p^2 |\langle \theta, \theta' \rangle| - 1 \right) \\
&\overset{(a)}{\le} \frac{4}{p^2} \left( \exp\left( p^4 \langle \theta, \theta' \rangle^2 \right) - 1 \right) \\
&\overset{(b)}{\le} \exp\left( \frac{\langle \theta, \theta' \rangle^2}{\varepsilon^4} \right) - 1,
\end{aligned}
$$

where (a) is due to the fact that $e^x - x \le e^{x^2}$ for all $x \ge 0$ and (b) follows from $p \ge 2$. $\qquad\square$

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

We will use the following lemma, originally stated in [DKK$^+$16], to prove Theorem 6.

**Lemma 6** ([Li17, Lemma A.5])**.** *Let $\mathcal{C}$ be a class of probability distributions. Suppose that for some fixed $N, \varepsilon, \gamma, \delta > 0$ there exists an algorithm that given $N$ independent samples from some $D \in \mathcal{C}$, of which up to an $\varepsilon$-fraction is corrupted, returns a list of $M$ distributions such that, with probability $1 - \delta/3$, there exists a $D'$ in the list with $\mathrm{d}_{\mathrm{TV}}(D', D) < \gamma$. Suppose furthermore that with probability $1 - \delta/3$, the distributions returned by this algorithm are all in some fixed set $\mathcal{M}$. Then there exists another algorithm, which given $O\big(N + \frac{1}{\varepsilon^2}\big(\log(|\mathcal{M}|) + \log\frac{1}{\delta}\big)\big)$ samples from $D$, an $\varepsilon$-fraction of which have been arbitrarily corrupted, returns a single distribution $D'$ so that with $1 - \delta$ probability $\mathrm{d}_{\mathrm{TV}}(D', D) < O(\gamma + \varepsilon)$.*

*Proof of Theorem 6.* Let $\mathcal{M}_A$ be the set of distributions defined as follows:

$$\mathcal{M}_A = \left\{ \mathcal{N}\left(\mu', I_d\right) : \|\mu'\|_0 \le m, \|\mu' - \mu\|_2 \le A \text{ and, for all } i, \mu'_i = k_i \frac{\varepsilon}{\sqrt{d}} \text{ for some } k_i \in \mathbb{Z} \right\}.$$

First, we will show that there exists a $\mathcal{N}\left(\mu', I_d\right) \in \mathcal{M}_A$ such that $\|\mu' - \mu\|_2 \le O(\varepsilon)$. Let $J \subseteq [d]$ be the set of indices of $m$ coordinates of $\mu$ with largest magnitude. Define $\mu'$ as a vector with coordinates $\mu'_j = \frac{\varepsilon}{\sqrt{d}} \left[\frac{\sqrt{d}\mu_j}{\varepsilon}\right] \mathbb{1}_J$, where $[\cdot]$ is the rounding operator. Then we have

$$\|\mu' - \mu\|_2^2 = \sum_{j \in J}(\mu_j - \mu'_j)^2 + \sum_{j \in J^C} \mu_j^2 \le \frac{\varepsilon^2 m}{d} + \sum_{j \in J^C} \mu_j^2. \tag{9}$$

The first term of the r.h.s. is at most $\varepsilon^2$; as for the second, it can be upper bounded as follows:

$$\sum_{j \in J^C} \mu_j^2 = \sum_{j \in J^C} |\mu_j|^{2-q} |\mu_j|^q \le \max_{u \in J^C} |\mu_u|^{2-q} \sum_{j \in J^C} |\mu_j|^q$$

$$\overset{(a)}{\le} \frac{\|\mu\|_q^{2-q} \|\mu\|_q^q}{(m+1)^{\frac{2-q}{q}}} = \frac{\|\mu\|_q^2}{(m+1)^{2/q-1}}$$

$$\le s^2 (m+1)^{1-2/q} \overset{(b)}{\le} \varepsilon^2,$$

where (a) uses the fact that $\max_{u \in J^C} |\mu_u|$ is the $(m+1)^{\text{th}}$ largest coordinate of $\mu$ and (b) is by the definition of $m$. Hence, we get $\|\mu' - \mu\|_2 \le \sqrt{2}\varepsilon$ (and, in particular, $\mu' \in \mathcal{M}_A$ for every $A \ge \sqrt{2}\varepsilon$).

It is easy to see that, for all $A > 0$, $|\mathcal{M}_A| \le \binom{d}{m}(A\sqrt{d}/\varepsilon)^m$: there are $\binom{d}{m}$ different ways to select the non-zero coordinates, and for each chosen set of non-zero coordinates there are at most $(A\sqrt{d}/\varepsilon)^m$ elements in $\mathcal{M}_A$.

Now, consider the following algorithm: First, use a naive pruning algorithm with $N$ samples as input, to output an approximation of $\mu$, denoted by $\mu_0$. Such a pruning is given and analyzed in [DKK$^+$16], which outputs $\mu_0$ such that $\|\mu_0 - \mu\|_2 \le B := O\big(\sqrt{d\log(N/\delta)}\big)$ with probability at least $1 - \delta$. Next, round each coordinate of $\mu_0$ to its nearest integer multiple of $\frac{\varepsilon}{\sqrt{d}}$, and output the set of distributions

$$\mathcal{M}' = \left\{ \mathcal{N}\left(\mu'', I\right) : \|\mu''\|_0 \le m, \|\mu'' - \mu_0\|_2 \le B \text{ and, for all } i, \mu''_i = k_i \frac{\varepsilon}{\sqrt{d}} \text{ for some } k_i \in \mathbb{Z} \right\}.$$

By the triangle inequality, with probability at least $1 - \delta$ we have $\mathcal{M}' \subseteq \mathcal{M}_{2B}$. Hence, by Lemma 6 there exists an algorithm that, with probability $1 - \delta$, outputs a $\mu'$ such that $\|\mu' - \mu\|_2 \le O(\varepsilon)$ and the number of samples required is

$$O\left(N + \frac{\log|\mathcal{M}_{2B}| + \log\frac{1}{\delta}}{\varepsilon^2}\right) = O\left(\frac{\log\binom{d}{m} + m\log\frac{d}{\varepsilon} + \log\frac{1}{\delta}}{\varepsilon^2}\right)$$

$$= O\left(\frac{m\log\frac{ed}{m} + m\log\frac{d}{\varepsilon} + \log\frac{1}{\delta}}{\varepsilon^2}\right).$$

This proves Theorem 6. $\qquad\qquad\square$

## 3.2 Testing in Sparse Linear Regression Model

In this section, we state the formal version of Theorem 3, and provide its proof.

**Theorem 7.** *Let $\gamma > 0$ be sufficiently small, $\varepsilon = \frac{\gamma}{C}$ for a sufficiently large $C$, and $s = d^{1-\delta}$ for some $\delta \in (0, 1)$. Let $(X_1, y_1), (X_2, y_2), \cdots, (X_n, y_n)$ be i.i.d. samples obtained from the sparse linear regression model described in* (3). *Moreover, suppose an $\varepsilon$-fraction of these $n$ samples are arbitrarily corrupted. Then, if there exists an algorithm that distinguishes between the cases $\|\theta\|_2 = 0$ and $\|\theta\|_2 \geq \gamma$ with probability greater than 2/3, we must have $n = \Omega\left(\min\left(s \log d, \frac{1}{\gamma^4}\right)\right)$.*

*Proof.* Given $(X_1, y_1), (X_2, y_2), \cdots, (X_n, y_n)$ as in the statement, let $X := (X_1, X_2, \cdots, X_n)$ and $Y := (y_1, y_2, \cdots, y_n)$. As noted earlier, while deriving a lower bound, it is enough to consider the weaker $\varepsilon$-Huber model for the corruption of samples. We define a hypothesis testing task compliant with the $\varepsilon$-Huber contamination model: a lower bound on the sample complexity of this hypothesis testing task will then constitute a lower bound for the robust testing in sparse linear regression model. Let $\Theta := \{\theta \in \{-\frac{1}{\sqrt{s}}, 0, \frac{1}{\sqrt{s}}\}^d : \|\theta\|_0 = s\}$ and $p := \frac{1-\varepsilon}{\varepsilon}$. Further, denote the probability distribution under the hypotheses $\mathcal{H}_0$ and $\mathcal{H}_1$ by $\mathbf{p}$ and $\mathbf{q}$, respectively. We define $\mathcal{H}_0$ and $\mathcal{H}_1$ as follows:

$$
\begin{aligned}
\mathcal{H}_0 : \quad & (X_i, y_i) \overset{\text{i.i.d.}}{\sim} \mathcal{N}(0, I_{d+1}) \\
\mathcal{H}_1 : \quad & \theta \sim \mathbf{u}_\Theta \\
& y_i \overset{\text{i.i.d.}}{\sim} \mathcal{N}(0, 1 + \gamma^2) \\
& \mathbf{q}(X_i \mid y_i, \theta) = \mathbf{q}_\theta(X_i \mid y_i) = (1 - \varepsilon)\mathcal{N}\left(\frac{y_i \gamma \theta}{1 + \gamma^2}, I - \frac{\gamma^2 \theta \theta^T}{1 + \gamma^2}\right) \\
& \qquad\qquad + \varepsilon \mathcal{N}\left(-\frac{(1 - \varepsilon)}{\varepsilon} \frac{y_i \gamma \theta}{(1 + \gamma^2)}, I - \frac{\gamma^2 \theta \theta^T}{1 + \gamma^2}\right)
\end{aligned}
\tag{10}
$$

Note that we can rewrite

$$
\mathbf{q}_\theta(X_i \mid y_i) = \mathbb{E}_b\left[\mathcal{N}\left(\frac{b y_i \gamma \theta}{1 + \gamma^2}, I - \frac{\gamma^2 \theta \theta^T}{1 + \gamma^2}\right)\right], \quad \text{where } b \sim (1 - \varepsilon)\delta_1 + \varepsilon \delta_{-p} ;
$$

in particular, $\mathbf{q}_\theta$ is an $\varepsilon$-Huber contaminated version of $\mathcal{N}(0, \Sigma_{\gamma\theta})$, where $\Sigma_{\gamma\theta} = \begin{bmatrix} I_d & \gamma\theta \\ \gamma\theta^T & 1 + \gamma^2 \end{bmatrix}$ and thus a valid distribution to consider in this problem. A natural approach to prove the impossibility of detection between $\mathcal{H}_0$ and $\mathcal{H}_1$ would be to try and show that $\chi^2(\mathbf{q} \| \mathbf{p}) = o(1)$. However, as previously discussed, this method does not yield a useful lower bound for this problem, due to low-probability events which cause $\chi^2(\mathbf{q} \| \mathbf{p})$ to blow up.

To circumvent this issue, we take recourse in an alternative approach, the *conditional second moment method* [RXZ19, WX18]. In this method, we first define a high-probability event $\mathcal{E}$ and then evaluate $\chi^2(\mathbf{q}^{\mathcal{E}} \| \mathbf{p})$, where $\mathbf{q}^{\mathcal{E}}$ is the distribution $\mathbf{q}$ conditioned on the event $\mathcal{E}$. The idea is that, by this conditioning, we can rule out the rare events that cause $\chi^2(\mathbf{q} \| \mathbf{p})$ to go to infinity. Indeed, suppose the event $\mathcal{E}$ is chosen so that $\mathbf{q}(\mathcal{E}) = o(1)$, and that we are able to show that $\chi^2(\mathbf{q}^{\mathcal{E}} \| \mathbf{p}) = o(1)$. This latter statement implies that $\mathrm{d}_{\mathrm{TV}}(\mathbf{q}^{\mathcal{E}}, \mathbf{p}) = o(1)$, and so, from the relation $\mathbf{q} = \mathbf{q}(\mathcal{E})\mathbf{q}^{\mathcal{E}} + \mathbf{q}(\mathcal{E}^C)\mathbf{q}^{\mathcal{E}^C}$ and the convexity of total variation distance, we have

$$
\mathrm{d}_{\mathrm{TV}}(\mathbf{q}, \mathbf{p}) \leq \mathbf{q}(\mathcal{E}) \mathrm{d}_{\mathrm{TV}}(\mathbf{q}^{\mathcal{E}}, \mathbf{p}) + \mathbf{q}(\mathcal{E}^C) \mathrm{d}_{\mathrm{TV}}(\mathbf{q}^{\mathcal{E}^C}, \mathbf{p}) \leq \mathrm{d}_{\mathrm{TV}}(\mathbf{q}^{\mathcal{E}}, \mathbf{p}) + \mathbf{q}(\mathcal{E}^C) = o(1),
$$

which proves the impossibility result.

We now implement this roadmap. For $\nu > 0$, we define the event $\mathcal{E}$ as follows:

$$
\mathcal{E} = \left\{ Y \in \mathbb{R}^d : \frac{\|Y\|_2^2}{n} \leq 2 + \nu \right\},
\tag{11}
$$

so that the conditional probability distribution $\mathbf{q}^{\mathcal{E}}$ is given by

$$
\mathbf{q}^{\mathcal{E}} = \mathbf{q}(X, Y \mid \mathcal{E}) = \frac{\mathbb{E}_\theta[\mathbf{q}_\theta(X, Y) \mathbb{1}_{\mathcal{E}}\{Y\}]}{\mathbf{q}(\mathcal{E})}.
\tag{12}
$$

We will later choose $\nu$ such that $\mathcal{E}$ is asymptotically a high-probability event, and for now focus on establishing that $\chi^2(\mathbf{q}^{\mathcal{E}} \parallel \mathbf{p}) = o(1)$. Note that $\chi^2(\mathbf{q}^{\mathcal{E}} \parallel \mathbf{p}) = \mathbb{E}_{\mathbf{p}}\left[\left(\frac{\mathbf{q}^{\mathcal{E}}}{\mathbf{p}}\right)^2\right] - 1$. Now, since $\frac{\mathbf{q}^{\mathcal{E}}}{\mathbf{p}} = \frac{\mathbb{E}_\theta[\mathbf{q}_\theta(X,Y)\mathbb{1}_{\mathcal{E}}\{Y\}]}{\mathbf{q}(\mathcal{E})\mathbf{p}(X,Y)}$, we have

$$\left(\frac{\mathbf{q}^{\mathcal{E}}}{\mathbf{p}}\right)^2 = \frac{1}{\mathbf{q}(\mathcal{E})^2}\mathbb{E}_{\theta,\theta'}\left[\frac{\mathbf{q}_\theta(X,Y)\mathbf{q}_{\theta'}(X,Y)}{\mathbf{p}(X,Y)^2}\mathbb{1}_{\mathcal{E}}\{Y\}\right],$$

and so

$$\mathbb{E}_{\mathbf{p}}\left[\left(\frac{\mathbf{q}^{\mathcal{E}}}{\mathbf{p}}\right)^2\right] = \frac{1}{\mathbf{q}(\mathcal{E})^2}\mathbb{E}_{\theta,\theta'}\left[\mathbb{E}_Y\left[\frac{\mathbf{q}(Y)^2}{\mathbf{p}(Y)^2}\mathbb{E}_X\left[\frac{\mathbf{q}_\theta(X\mid Y)\mathbf{q}_{\theta'}(X\mid Y)}{\mathbf{p}(X)^2}\right]\mathbb{1}_{\mathcal{E}}\{Y\}\right]\right],$$

where $Y \sim \mathcal{N}(0,1)^n$ and $X \sim \mathcal{N}(0,I_d)^n$. Since $\mathcal{E}$ is a high-probability event, we have $\mathbf{q}(\mathcal{E}) = 1 - o(1)$. Thus,

$$\mathbb{E}_{\mathbf{p}}\left[\left(\frac{\mathbf{q}^{\mathcal{E}}}{\mathbf{p}}\right)^2\right] = (1+o(1))\mathbb{E}_{\theta,\theta'}\left[\underbrace{\mathbb{E}_Y\left[\frac{\mathbf{q}(Y)^2}{\mathbf{p}(Y)^2}\mathbb{E}_X\left[\frac{\mathbf{q}_\theta(X\mid Y)\mathbf{q}_{\theta'}(X\mid Y)}{\mathbf{p}(X)^2}\right]\mathbb{1}_{\mathcal{E}}\{Y\}\right]}_{:=Z}\right]. \quad (13)$$

We will evaluate $\mathbb{E}_{\theta,\theta'}[Z]$ by splitting it into two cases depending on the value of $|\langle\theta,\theta'\rangle|$. Let $\tau := \frac{\varepsilon^2}{4e\gamma^2 \log d}$ (the choice of value for this threshold will become clear in the course of the argument).

**Case 1:** $|\langle\theta,\theta'\rangle| \leq \tau$. In this case we drop the $\mathbb{1}_{\mathcal{E}}\{Y\}$ term in the expression for $Z$, simply upper bounding it by 1. Note that in the absence of $\mathbb{1}_{\mathcal{E}}\{Y\}$ term, $Z$ becomes a product of expectations, enabling the following simplification:

$$\mathbb{E}_{\theta,\theta'}[Z\mathbb{1}\{|\langle\theta,\theta'\rangle| \leq \tau\}]$$
$$\leq \mathbb{E}_{\theta,\theta'}\left[\left(\mathbb{E}_y\left[\frac{\mathbf{q}(y)^2}{\mathbf{p}(y)^2}\mathbb{E}_x\left[\frac{\mathbf{q}_\theta(x\mid y)\mathbf{q}_{\theta'}(x\mid y)}{\mathbf{p}(x)^2}\right]\right]\right)^n\mathbb{1}\{|\langle\theta,\theta'\rangle| \leq \tau\}\right], \quad (14)$$

where $y \sim \mathcal{N}(0,1)$ and $x \sim \mathcal{N}(0,I_d)$. By substituting for $\mathbf{q}_\theta$ from (10) and using Corollary 7, we get

$$\mathbb{E}_x\left[\frac{\mathbf{q}_\theta(x\mid y)\mathbf{q}_{\theta'}(x\mid y)}{\mathbf{p}(x)^2}\right] \leq \frac{1}{\sqrt{1 - \gamma^4 \langle\theta,\theta'\rangle^2}}\mathbb{E}_{b,b'}\left[\exp\left(\gamma^2 y^2 bb' \langle\theta,\theta'\rangle\right)\right]. \quad (15)$$

Note that $\mathbb{E}[b] = 0$ and, for $\varepsilon \in (0,1/2]$, $|\mathbb{E}_b[b^\ell]| = |(1-\varepsilon) + (-1)^\ell p^\ell \varepsilon| = (1-\varepsilon)|1 + (-1)^\ell p^{\ell-1}| \leq 2p^{\ell-1}$. Therefore,

$$\mathbb{E}_{b,b'}\left[\exp\left(\gamma^2 y^2 bb' \langle\theta,\theta'\rangle\right)\right] = \mathbb{E}_{b,b'}\left[1 + \sum_{\ell=1}^{\infty} \frac{\gamma^{2\ell}y^{2\ell}b^\ell b'^\ell \langle\theta,\theta'\rangle^\ell}{\ell!}\right]$$
$$\leq 1 + \sum_{\ell=2}^{\infty} \frac{\gamma^{2\ell}y^{2\ell}\mathbb{E}_b[b^\ell]^2|\langle\theta,\theta'\rangle|^\ell}{\ell!}$$
$$\leq 1 + \frac{4}{p^2}\sum_{\ell=2}^{\infty} \frac{\gamma^{2\ell}y^{2\ell}p^{2\ell}|\langle\theta,\theta'\rangle|^\ell}{\ell!}.$$

By monotone convergence theorem, it then follows that

$$\mathbb{E}_y\left[\frac{\mathbf{q}(y)^2}{\mathbf{p}(y)^2}\mathbb{E}_{b,b'}\left[\exp\left(\gamma^2 y^2 bb' \langle\theta,\theta'\rangle\right)\right]\right]$$
$$\leq \mathbb{E}_y\left[\frac{\mathbf{q}(y)^2}{\mathbf{p}(y)^2}\right] + \frac{4}{p^2}\sum_{\ell=2}^{\infty} \frac{\gamma^{2\ell}p^{2\ell}|\langle\theta,\theta'\rangle|^\ell}{\ell!}\mathbb{E}_y\left[\frac{\mathbf{q}(y)^2}{\mathbf{p}(y)^2}y^{2\ell}\right].$$

Now, we use Lemma 3 and the fact that $\ell! \geq \left(\frac{\ell}{e}\right)^{\ell}$. Recalling that we chose $\tau = \frac{\varepsilon^2}{4e\gamma^2 \log d}$, we get

$$
\mathbb{E}_y \left[ \frac{\mathbf{q}(y)^2}{\mathbf{p}(y)^2} \mathbb{E}_{b,b'} \left[ \exp\left(\gamma^2 y^2 bb' \langle \theta, \theta' \rangle\right) \right] \right] \leq \frac{1}{\sqrt{1-\gamma^4}} \left( 1 + \frac{4c}{p^2} \sum_{\ell=2}^{\infty} \gamma^{2\ell} 4^{\ell} e^{\ell} p^{2\ell} |\langle \theta, \theta' \rangle|^{\ell} \right)
$$

$$
= \frac{1}{\sqrt{1-\gamma^4}} \left( 1 + \frac{64c}{p^2} \frac{\gamma^4 p^4 e^2 |\langle \theta, \theta' \rangle|^2}{(1 - \gamma^2 p^2 4e |\langle \theta, \theta' \rangle|)} \right)
$$

$$
\leq \frac{1}{\sqrt{1-\gamma^4}} \left( 1 + \frac{64c\gamma^4 p^2 e^2 |\langle \theta, \theta' \rangle|^2}{1 - \frac{1}{\log d}} \right).
$$
(16)

where $c > 0$ is the constant from Lemma 3. By (15), and (16) and for large enough $d$ we have

$$
\mathbb{E}_y \left[ \frac{\mathbf{q}(y)^2}{\mathbf{p}(y)^2} \mathbb{E}_x \left[ \frac{\mathbf{q}_\theta(x \mid y)\mathbf{q}_{\theta'}(x \mid y)}{\mathbf{p}(x)^2} \right] \right]
$$

$$
\leq \frac{1}{\sqrt{1-\gamma^4}} \frac{1}{\sqrt{1 - \gamma^4 \langle \theta, \theta' \rangle^2}} \left( 1 + \frac{64c\gamma^4 p^2 e^2 |\langle \theta, \theta' \rangle|^2}{1 - \frac{1}{\log d}} \right)
$$

$$
\leq \exp\left(\gamma^4\right) \exp\left(\gamma^4 \langle \theta, \theta' \rangle^2\right) \exp\left( \frac{128 e^2 c \cdot \gamma^4 \langle \theta, \theta' \rangle^2}{\varepsilon^2} \right).
$$

Substituting in (14), we get, for some absolute constant $C > 0$ (which one can take to be $C := 256e^2 c$),

$$
\mathbb{E}_{\theta,\theta'}[Z \mathbb{1}\{|\langle \theta, \theta' \rangle| \leq \tau\}] \leq \exp\left(n\gamma^4\right) \mathbb{E}_{\theta,\theta'} \left[ \exp\left( C \frac{n\gamma^4 \langle \theta, \theta' \rangle^2}{\varepsilon^2} \right) \mathbb{1}\{|\langle \theta, \theta' \rangle| \leq \tau\} \right]
$$

$$
\leq \exp\left(n\gamma^4\right) \mathbb{E}_{\theta,\theta'} \left[ \exp\left( C \frac{n\gamma^4 \langle \theta, \theta' \rangle^2}{\varepsilon^2} \right) \right].
$$

Let $\beta := \sqrt{s}\theta$ and $\beta' := \sqrt{s}\theta'$. Then, we can rewrite the abolve inequality as

$$
\mathbb{E}_{\theta,\theta'}[Z \mathbb{1}\{|\langle \theta, \theta' \rangle| \leq \tau\}] \leq \exp\left(n\gamma^4\right) \mathbb{E}_{\beta,\beta'} \left[ \exp\left( C \frac{n\gamma^4 \langle \beta, \beta' \rangle^2}{\varepsilon^2 s^2} \right) \right].
$$

From there, we can use an argument similar to that in the proof of Theorem 4. It suffices to evaluate the expectation by fixing $\beta'$, say to $\beta_0$, where $\beta_0 = (\underbrace{1, 1, \cdots, 1}_{s}, 0, 0, \cdots, 0)$. The random variable $G := \langle \beta, \beta_0 \rangle$ is then a symmetric random walk with Hypergeometric$(d, s, s)$ stopping time. By Lemma 2, for $n = o\left( \min\left( \frac{\varepsilon^2 s}{\gamma^4} \log \frac{ed}{s}, \frac{1}{\gamma^4} \right) \right)$, we have

$$
\mathbb{E}_{\theta,\theta'}[Z \mathbb{1}\{|\langle \theta, \theta' \rangle| \leq \tau\}] \leq \exp\left(n\gamma^4\right) \mathbb{E}_{\theta,\theta'} \left[ \exp\left( C \frac{n\gamma^4 G^2}{\varepsilon^2 s^2} \right) \right] = 1 + o(1), \quad (17)
$$

which concludes the analysis of this case.

**Case 2:** $|\langle \theta, \theta' \rangle| > \tau$. From (13) we can rewrite

$$
\mathbb{E}_{\theta,\theta'}[Z \mathbb{1}\{|\langle \theta, \theta' \rangle| > \tau\}]
$$

$$
= \mathbb{E}_{\theta,\theta'} \left[ \mathbb{E}_Y \left[ \frac{\mathbf{q}(Y)^2}{\mathbf{p}(Y)^2} \prod_{i=1}^{n} \mathbb{E}_{x_i} \left[ \frac{\mathbf{q}_\theta(x_i \mid y_i)\mathbf{q}_{\theta'}(x_i \mid y_i)}{\mathbf{p}(x_i)^2} \right] \mathbb{1}_{\mathcal{E}}\{Y\} \right] \mathbb{1}\{|\langle \theta, \theta' \rangle| > \tau\} \right].
$$
(18)

To proceed further, we will rely on the following technical lemma, a corollary of Lemma 1 whose proof we defer to the end of the section:

**Lemma 7.** *Let $\phi_{\mu,\Sigma}$ denote the density function of $\mathcal{N}(\mu, \Sigma)$. Then, for $\mu_1 = \frac{\gamma b \theta y}{1+\gamma^2}$,*
*$\mu_2 = \frac{\gamma b' \theta' y}{1+\gamma^2}$, $\Sigma_1 = I - \frac{\gamma^2 \theta \theta^T}{1+\gamma^2}$, and $\Sigma_2 = I - \frac{\gamma^2 \theta' \theta'^T}{1+\gamma^2}$, we have*

$$\mathbb{E}_{X \sim \phi_{0,I}} \left[ \frac{\phi_{\mu_1, \Sigma_1}(X) \phi_{\mu_2, \Sigma_2}(X)}{\phi_{0,I}^2(X)} \right] \leq \frac{1}{\sqrt{1 - \gamma^4 \langle \theta, \theta' \rangle^2}} \exp\left( \gamma^2 y^2 bb' \langle \theta, \theta' \rangle \right).$$

Invoking Lemma 7, we then can bound the inner expectations as

$$\mathbb{E}_{x_i} \left[ \frac{\mathbf{q}_\theta(x_i \mid y_i) \mathbf{q}_{\theta'}(x_i \mid y_i)}{\mathbf{p}(x_i)^2} \right] \leq \frac{1}{\sqrt{1 - \gamma^4 \langle \theta, \theta' \rangle^2}} \mathbb{E}_{b_i, b_i'} \left[ \exp\left( \gamma^2 y_i^2 b_i b_i' \langle \theta, \theta' \rangle \right) \right]. \quad (19)$$

Let $b = (b_1, b_2, \cdots, b_n)$ and $b' = (b_1', b_2', \cdots, b_n')$ be i.i.d. random variables, where $b_i, b_i' \sim (1 - \varepsilon)\delta_1 + \varepsilon\delta_{-p}$. Then

$$\prod_{i=1}^n \mathbb{E}_{b_i, b_i'} \left[ \exp\left( \gamma^2 y_i^2 b_i b_i' \langle \theta, \theta' \rangle \right) \right] = \mathbb{E}_{b,b'} \left[ \exp\left( \gamma^2 \left( \sum_{i=1}^n b_i b_i' y_i^2 \right) \langle \theta, \theta' \rangle \right) \right]$$

$$\leq \mathbb{E}_{b,b'} \left[ \exp\left( \gamma^2 \left| \sum_{i=1}^n b_i b_i' y_i^2 \right| | \langle \theta, \theta' \rangle | \right) \right]$$

$$\leq \exp\left( \gamma^2 p^2 \|Y\|_2^2 | \langle \theta, \theta' \rangle | \right). \quad (20)$$

Using (19) and (20) we get

$$\mathbb{E}_Y \left[ \frac{\mathbf{q}(Y)^2}{\mathbf{p}(Y)^2} \prod_{i=1}^n \mathbb{E}_{X_i} \left[ \frac{\mathbf{q}_\theta(X_i \mid y_i) \mathbf{q}_{\theta'}(X_i \mid y_i)}{\mathbf{p}(X_i)^2} \right] \mathbb{1}_{\mathcal{E}}\{Y\} \right]$$

$$\leq \frac{1}{(1 - \gamma^4)^{n/2}} \mathbb{E}_Y \left[ \frac{\mathbf{q}(Y)^2}{\mathbf{p}(Y)^2} \exp\left( \gamma^2 p^2 \|Y\|_2^2 | \langle \theta, \theta' \rangle | \right) \mathbb{1}_{\mathcal{E}}\{Y\} \right],$$

which by using the definition of $\mathcal{E}$ in (11) can be bounded as

$$\mathbb{E}_Y \left[ \frac{\mathbf{q}(Y)^2}{\mathbf{p}(Y)^2} \prod_{i=1}^n \mathbb{E}_{X_i} \left[ \frac{\mathbf{q}_\theta(X_i \mid y_i) \mathbf{q}_{\theta'}(X_i \mid y_i)}{\mathbf{p}(X_i)^2} \right] \mathbb{1}_{\mathcal{E}}\{Y\} \right]$$

$$\leq \frac{\exp\left( \gamma^2 p^2 n(2 + \nu) | \langle \theta, \theta' \rangle | \right)}{(1 - \gamma^4)^{n/2}} \mathbb{E}_Y \left[ \frac{\mathbf{q}(Y)^2}{\mathbf{p}(Y)^2} \right]$$

$$= \frac{\exp\left( \gamma^2 p^2 n(2 + \nu) | \langle \theta, \theta' \rangle | \right)}{(1 - \gamma^4)^n}.$$

Recall that $| \langle \theta, \theta' \rangle | \leq \frac{H}{s}$, where $H \sim \text{Hypergeometric}(d, s, s)$. Therefore, plugging the above equation in (18) yields, letting $\lambda := \frac{\gamma^2 p^2 n(2+\nu)}{s}$,

$$\mathbb{E}_{\theta, \theta'}[Z \mathbb{1}\{ | \langle \theta, \theta' \rangle | > \tau \}] \quad (21)$$

$$\leq \frac{1}{(1 - \gamma^4)^n} \mathbb{E}_{\theta, \theta'} \left[ \exp\left( \frac{\gamma^2 p^2 n(2 + \nu)}{s} s | \langle \theta, \theta' \rangle | \right) \mathbb{1}\{ | \langle \theta, \theta' \rangle | > \tau \} \right]$$

$$\leq \frac{1}{(1 - \gamma^4)^n} \mathbb{E}_H[\exp(\lambda H) \mathbb{1}\{H > s\tau\}]$$

$$\leq \exp\left( 2n\gamma^4 \right) \mathbb{E}_H[\exp(\lambda H) \mathbb{1}\{H > s\tau\}]. \quad (22)$$

We use Lemma 4 to evaluate the r.h.s. in the above equation as follows:

$$\mathbb{E}_H[\exp(\lambda H)\mathbb{1}\{H > s\tau\}] = \sum_{h=\lceil s\tau \rceil}^{s} \exp(\lambda h) \Pr[H = h]$$

$$\leq \sum_{h=\lceil s\tau \rceil}^{s} \exp(\lambda h) \left(\frac{es^2}{h(d-s+1)}\right)^h$$

$$= \sum_{h=\lceil s\tau \rceil}^{s} \exp\left(\lambda h - h\log\frac{h(d-s+1)}{es^2}\right)$$

$$\leq \sum_{h=\lceil s\tau \rceil}^{\infty} \exp\left(-h\left(\log\frac{\tau(d-s+1)}{es} - \lambda\right)\right)$$

$$= \frac{\exp\left(-s\tau\underbrace{\left(\log\frac{\tau(d-s+1)}{es} - \lambda\right)}_{:=T}\right)}{1 - \exp\left(-\left(\log\frac{\tau(d-s+1)}{es} - \lambda\right)\right)}. \qquad (23)$$

Substituting for $\lambda$ in $T$, we get

$$T = \log\frac{\tau(d-s+1)}{es} - \frac{\gamma^2 n(2+\nu)(1-\varepsilon)^2}{\varepsilon^2 s}.$$

Recall that we already have fixed $\tau = \frac{\varepsilon^2}{4e\gamma^2 \log d}$, but kept $\nu$ (which appears in the definition of $\mathcal{E}$, in (11)) as a free parameter. Set $\nu := \frac{\log\log d}{n}$. From the theorem statement, we have $\varepsilon = \frac{\gamma}{C}$ and $s = d^{1-\delta}$, and thus,

$$T = \log\left(\frac{(d-s+1)\varepsilon^2}{4e^2\gamma^2 s \log d}\right) - \frac{\gamma^2(2n + \log\log d)(1-\varepsilon)^2}{\varepsilon^2 s} = \Theta\left(\log d - \log\log d - \frac{n}{s}\right).$$

If $n = o(s\log d)$, this implies $T = \omega(1)$. Since $s\tau$ is an increasing function of dimension, (23) then further gives that

$$\mathbb{E}_H[\exp(\lambda H)\mathbb{1}\{H > s\tau\}] = o(1)$$

Therefore, by (21), if $n = o\left(\min\left(s\log d, \frac{1}{\gamma^4}\right)\right)$,

$$\mathbb{E}_{\theta,\theta'}[Z\mathbb{1}\{|\langle \theta, \theta'\rangle| > \tau\}] = o(1).$$

which concludes the analysis of the second case.

Plugging the bounds derived in Case 1 and Case 2 in (13) imply that $\mathbb{E}_{\mathbf{P}}\left[\left(\frac{\mathbf{q}^{\mathcal{E}}}{\mathbf{P}}\right)^2\right] = 1 + o(1)$ whenever $n = o\left(\min\left(s\log d, 1/\gamma^4\right)\right)$, and so

$$\chi^2(\mathbf{q}^{\mathcal{E}} \| \mathbf{p}) = o(1).$$

It remains to show that $\mathcal{E}$ is a high probability event for the chosen $\nu$, i.e., $\nu = \frac{\log\log d}{n}$. Let $Q_{\chi_n^2}$ denote the $Q$-function of $\chi^2$ distribution with $n$ degrees of freedom. Then, by the definition of $\mathcal{E}$,

$$\mathbf{q}\left(\mathcal{E}^C\right) = Q_{\chi_n^2}(n(2+\nu))$$

$$\overset{(a)}{\leq} Q_{\chi_n^2}(n(1 + \sqrt{\nu} + \nu/2))$$

$$\overset{(b)}{\leq} \exp\left(-\frac{1}{4}n\nu\right) = \exp\left(-\frac{1}{4}\log\log d\right) = o(1),$$

where (a) is due to the fact that $1 + u \geq \sqrt{u} + u/2$ for all $u \geq 0$ and (b) is due to standard concentration inequalities for $\chi_n^2$ random variables.

This was the last piece missing: by the argument outlined at the beginning of this proof, we conclude that
$$d_{\mathrm{TV}}(\mathbf{p}, \mathbf{q}) = o(1)$$
that is, that it is impossible to distinguish between $\mathcal{H}_0$ and $\mathcal{H}_1$ with constant probability, as long as $n = o\left(\min\left(s \log d, \frac{1}{\gamma^4}\right)\right)$. This concludes the proof of Theorem 7. $\qquad\square$

To conclude, it only remains to prove the technical lemma we invoked in the above proof, Lemma 7.

*Proof of Lemma 7.* Plugging in the values of $\mu_1, \mu_2, \Sigma_1, \Sigma_2$ in Lemma 1, we can expand

$$\mathbb{E}_{X \sim \phi_{0,I}}\left[\frac{\phi_{\mu_1,\Sigma_1}(X)\phi_{\mu_2,\Sigma_2}(X)}{\phi_{0,I}^2(X)}\right] = \frac{1 + \gamma^2}{\det(A)^{1/2}} \exp\left(\frac{1}{2}U\right), \tag{24}$$

where $A = I + \gamma^2(\theta\theta^T + \theta'\theta'^T)$ and

$$U = \gamma^2 y^2 (b\theta + b'\theta')^T A^{-1}(b\theta + b'\theta') - b^2\gamma^2 y^2 + \frac{b^2\gamma^4 y^2}{1 + \gamma^2} - b'^2\gamma^2 y^2 + \frac{b'^2\gamma^4 y^2}{1 + \gamma^2}.$$

Using the matrix inversion identity, we then have

$$\begin{aligned}
A^{-1} &= \left(I + \gamma^2[\theta,\theta'][\theta,\theta']^T\right)^{-1} \\
&= I - \gamma^2[\theta,\theta']\begin{bmatrix} 1 + \gamma^2 & \gamma^2\langle\theta,\theta'\rangle \\ \gamma^2\langle\theta,\theta'\rangle & 1 + \gamma^2 \end{bmatrix}^{-1}[\theta,\theta']^T \\
&= I - \frac{\gamma^2}{(1+\gamma^2)^2 - \gamma^4\langle\theta,\theta'\rangle^2}\Big(\underbrace{(1+\gamma^2)(\theta\theta^T + \theta'\theta'^T) - \gamma^2\langle\theta,\theta'\rangle(\theta'\theta^T + \theta\theta'^T)}_{:=V}\Big).
\end{aligned}$$

Thus,

$$U = \gamma^2 y^2 \|b\theta + b'\theta'\|^2 - b^2\gamma^2 y^2 - b'^2\gamma^2 y^2 - \frac{\gamma^4 y^2\left((b\theta + b'\theta')^T V(b\theta + b'\theta')\right)}{(1+\gamma^2)^2 - \gamma^4\langle\theta,\theta'\rangle^2} + \frac{b^2\gamma^4 y^2}{1 + \gamma^2} + \frac{b'^2\gamma^4 y^2}{1 + \gamma^2}.$$

For sufficiently small $\gamma$, i.e., smaller than some absolute constant $\gamma_0 > 0$ (recalling that $\theta, \theta'$ are unit vectors, and so $V$ is bounded), we get

$$U \leq 2\gamma^2 y^2 bb'\langle\theta,\theta'\rangle. \tag{25}$$

Using the matrix determinant identity, we have

$$\det(A) = \det(I + \gamma^2[\theta,\theta'][\theta,\theta']^T) = \det\left(I + \gamma^2\begin{bmatrix} 1 & \langle\theta,\theta'\rangle \\ \langle\theta,\theta'\rangle & 1 \end{bmatrix}\right) = (1+\gamma^2)^2 - \gamma^4\langle\theta,\theta'\rangle^2.$$

Thus,

$$\frac{1 + \gamma^2}{\det(A)^{1/2}} = \frac{1}{\sqrt{1 - \frac{\gamma^4\langle\theta,\theta'\rangle^2}{(1+\gamma^2)^2}}} \leq \frac{1}{\sqrt{1 - \gamma^4\langle\theta,\theta'\rangle^2}}. \tag{26}$$

Eqs. (24), (25) and (26) together prove the lemma. $\qquad\square$

## 4 Discussion and Future Work

In this section, we discuss some of the limitations of our results, which we believe are ground for possible future work. Firstly, we suspect that our lower bounds are not tight with respect to the parameters $\gamma$ and $\varepsilon$. The dependence on $\gamma$ and $\varepsilon$ in Theorems 4 and 5, is $\Omega(\varepsilon^4/\gamma^4)$. While the exact dependence on these parameters is still an open problem even for robust Gaussian mean testing (non-sparse), we conjecture that the actual dependence on these parameters should scale as $\varepsilon^2/\gamma^4$, and hence believe that our dependence on these parameters is suboptimal. In the current proof, the bottleneck appears while obtaining an upper bound for the chi-squared correlation between two

Huber-contaminated distributions, and we suspect that more advanced techniques might be required to get the right dependence on $\gamma$ and $\varepsilon$.

Furthermore, we restricted ourselves to the case when the covariance of the Gaussian distributions under consideration are identity matrices ("spherical Gaussians"); of course, handling the case of arbitrary covariances is a natural and important question. We believe that deriving the sample complexity in the case of non-identity (and unknown) covariance matrices would require significant additional effort, as well as new techniques and ideas. It is worth pointing out that we are not aware of any work addressing the sample complexity results for non-identity covariances even in the non-robust (but sparse) setting.