# OpenReview forum: "Robust Testing in High-Dimensional Sparse Models"
_NeurIPS.cc/2022/Conference — NeurIPS 2022 Accept_

### Official Review · Reviewer_2TQX · 2022-07-10

**Rating:** 3
**Confidence:** 2
**Soundness:** 3 good
**Presentation:** 1 poor
**Contribution:** 2 fair

**Summary:**

This paper is interested in determining whether the norm of the mean (supposed to be sparse) of a Gaussian distribution is zero or not based on samples drawn from this distribution. The difficulty of the setting considered comes from the addition of a robustness condition which imposes that a fraction of the data available for the test is corrupted. The authors are interested in determining the minimum number of samples (sample complexity) needed to perform such a task. The authors hope through the study to show the impact of robustness on pre-existing results on the sample complexity of the testing the norm of sparse gaussian vectors. In particular, they show that the robustness constraints significantly increase the complexity of the test. It should be noted that an additional study to test the norm of a regression vector in a sparse linear regression context is also proposed and theoretically analyzed with the same consequences.

**Questions:**

1- In line 43 , $p_{0}^n$ seems not to be defined

2- Line 108 "we note that for in"

3-Line 169 $\tilde{\mathcal{D}}$ seems not defined

4- The single MGF seems not defined

5- It would be interesting to have a bunch of experiments to illustrate the theory. In particular, could the authors provide a set of baselines algorithms and their sample complexity and how the proposed sample complexity behaves with respect to this empirical sample complexities?

6- The presentation can be improved by summarizing the main ingredients of the proofs more shortly and focusing more on interpretation, and visualization of the main theoretical results to increase readability, and make the paper more understandable.

7- Do they exist some robustness algorithms that could reach the proposed sample complexity?

**Limitations:**

1- The paper didn't discuss really the limitations or openings of the works.

2- The motivation of such a study notably for application is lacking to some extent.

3- Experiments are not provided and could be an important asset to support the theory.

4- Some algorithms that reach the sample complexity would be more than welcome.

**Strengths And Weaknesses:**

*Strengths*

1- The paper seems technically sound with original technics to derive the sample complexity.

2- The sample complexity derived makes some sense and provides some intuition into statistical tests under robustness conditions

*Weaknesses*

1- The work needs to be better motivated by giving concrete examples of the usefulness of testing the norm of a high dimensional sparse Gaussian vector.

2- Some experiments would have been welcome. Do some robust statistical tests exist to show empirically what are their sample complexities and how it relates to the derived theoretical sample complexity?

3- The structure of the paper could be improved by providing more opening and conclusions to the work, by insisting more on insights drawn more than the technical proofs which could be deferred to the supplementary materials. Furthermore, a notation section would be welcome to introduce some notation conventions.

---

> ### Author Response · Authors · 2022-08-01
> **Response to the review by 2TQX**
>
> We thank the reviewer for their time and comments. We address the specific questions below, and hope that the reviewer will reconsider their score in light of our answers.
>
> **Responses to Questions**
>
> 1-4: We thank the reviewer for catching these errors. In the paper, $\mathbf{p}_0$ refers to the reference distribution (simple null hypothesis, in our case the standard Gaussian). $\tilde{\mathcal{D}}$ is a typo--it should have been $\tilde{\mathcal{D}}_s$. We will correct these in the final version.
>
> 5: We thank the reviewer for this question. We acknowledge that this would be a very valuable state-of-knowledge (SoK) or survey paper. However, considering the fact that the focus of our paper is on deriving lower bound, we think that these experiments would help little in illustrating the results. Indeed, our lower bounds are achieved by already known algorithms for the corresponding learning problems (hence the message that “robust testing is as hard as learning”), and we believe that implementing the algorithms from these previous papers to show how they perform empirically on this different learning task would distract from our own results and focus.
>
> 6: Thank you for this suggestion. We will try our best to include more interpretation and visualization of our results in the final version (specifically, as a figure/phase diagram showing the various regimes of the sample complexity), subject to space constraints.
>
> 7: Indeed: agnostic hypothesis selection via tournaments algorithm in [Li17] and Theorem 2 of our paper achieve the sample complexity of our lower bound for robust sparse Gaussian mean testing. The same algorithm can be used for robust testing in sparse linear regression model in view of the results in [LSLC20, Theorem 2.1] which states that any algorithm for robust sparse mean estimation can be used for robust sparse linear regression with a polylog($1/\gamma$) increase in the sample complexity.  However, these algorithms are computationally inefficient. The best-known computationally efficient algorithm takes $O(s^2)$ samples [BDLS17], and this is conjectured to be inherent to these problems (we discuss these works, and the corresponding evidence backing up this conjecture, in Section 1.3).
>
>     [LSLC20] Liu Liu, Yanyao Shen, Tianyang Li, and Constantine Caramanis. High dimensional robust sparse regression. PMLR 108:411-421, 2020.
>
> **Response to Limitations**:
>
> 1: This is a great point: we will include a discussion of limitations and future work in the final version, and outline some of them now. One of the main limitations of our work is that the sample complexity is not tight in parameters $\varepsilon$ and $\gamma$ (see responses to reviewers jAix and PDuZ and for a discussion of this dependence). Importantly, to the best of our knowledge this dependence is unknown even in the non-sparse case. Hence, this is also one of the main future directions to explore; we believe that new techniques will be required to address it.
>     Another line to pursue further would be to find the sample complexities of these problems when the covariance of the Gaussian is non-identity (see response to reviewer jAix).  We are not aware of sample complexity results even in the non-robust (but sparse) setting where the covariance matrix is non-identity and unknown.
>
> 2: We thank the reviewer for pointing out this, and elaborate on some of the motivation below. Testing the norm of the mean vector is of fundamental interest to signal processing and statistics community, where it falls under the general name of Gaussian Location Model (GLM), and has been the subject of a long line of works. Furthermore, sparse signals, as a model and class of distributions, play a significant role in many applications – whenever applicable, they significantly reduce the sample and computational complexity. A few of the most relevant or initial papers in that respect include [Ing97], [DJ04]; see also, e.g., [ITV10] for the sparse regression question, or [CCC+19]. The setting in our paper corresponds to robustly detecting the presence of a high-dimensional sparse signal with high confidence, thus combining the practical motivations of robust statistics (data is routinely noisy, and modeling assumptions never exactly hold) to the motivations for sparse testing and sparse regression. We will detail these points further and add the corresponding citations in the final version of this paper.
>
>     [Ing97] Ingster, Y.I. (1997). Some problems of hypothesis testing leading to infinitely divisible distributions. Math. Methods Statist. 6 47–49.
>     [DJ04] Donoho, D.L. and Jin, J. (2004). Higher criticism for detecting sparse heterogeneous mixtures. Ann. Statist. 32 962–994.
>
> 3-4: We refer the reviewer to our answers to questions 5 and 7 for these two comments. We would be happy to elaborate further on any of the points raised above; we hope our answers address the reviewer’s concerns, and will lead them to re-evaluate their score.

---

### Official Review · Reviewer_PDuZ · 2022-07-11

**Rating:** 6
**Confidence:** 3
**Soundness:** 4 excellent
**Presentation:** 4 excellent
**Contribution:** 3 good

**Summary:**

The submission considers the well-known question of testing whether the mean of a $d$-dimensional identity-covariance Gaussian is zero or a sparse vector $\gamma$-far from zero, and the related question of testing whether the coefficient vector in an instance of sparse linear regression is zero or far from zero.

In this review, we will focus on the usual $L_0$ notion of sparsity, though their results for mean testing also extend to any $L_q$-based sparsity for $0 < q < 2$. For mean testing, classically it is known that the optimal sample complexity, up to log factors, scales as $\min(s,\sqrt{d})/\gamma^2$ where $s$ is the sparsity of the alternative hypothesis, and a qualitatively similar picture holds for SLR testing. In particular, there is a phase transition at $s = \sqrt{d}$ such that above this, testing is easier than estimation and, more specifically, the sample complexity does not grow with $s$ and at worst scales as $\sqrt{d}/\gamma^2$.

The present work considers a twist on this setup where some small constant fraction $\epsilon$ of the samples that the tester sees have been adversarially corrupted. They show that in this model, the abovementioned phase transition does not occur, and testing is no easier than estimation regardless of $s$. That is, they show that for any constant $\epsilon,\gamma$, the sample complexity for *robust* Gaussian mean testing scales linearly in $s$, and similarly for robust SLR testing. Note that this result was already known in the case of $s = d$ [DKS17].

The proofs are via the Ingster-Suslina method, which amounts to designing a mixture over alternatives and controlling the tails of the pairwise correlation between the distributions over samples under two random alternatives relative to the null hypothesis. In the robust setting, there is the additional freedom of choosing how the adversary corrupts the data: for mean testing, e.g., they consider an adversary (the same as in [DKS17]) that, for every sample, with probability $\epsilon$ replaces the sample with a draw from another Gaussian such that the mixture of the "clean Gaussian" and this Gaussian has zero mean.

**Questions:**

*Questions*:
- Do the lower bound techniques get the right $\gamma$ scaling as well? If not, where are the bottlenecks in the argument that preclude this?
- Is Brennan-Bresler's lower bound for robust SLR also a lower bound against robust SLR testing?

*Minor typos*:
- P. 3 Line 98: no longer present*s*
- P. 4 Line 124: "for some $\delta$" -> "for any $\delta$"?
- P. 4 Line 135: semicolon should be comma, similarly on P. 5 Line 185
- P. 4 Line 138: we note that in order
- P. 5 Line 186: *a* suitable event


**Limitations:**

The authors have adequately addressed the limitations, and I don't see any potential negative societal impact from this work.

**Strengths And Weaknesses:**

*Strengths*:
- It is nice that this paper gives an essentially complete answer to the question they set out to solve, up to log factors and dependence on $\epsilon,\gamma$.
- While the techniques employed are all fairly standard, they are integrated in a clean and modular fashion.
- Conceptual novelty: while there has been a lot of work on the estimation version of the problems this paper studies, as far as I know this is the first to propose studying robust SLR testing, even for $s = d$.

*Weaknesses*:
- For the mean testing result, a lot of the technical heavy lifting, namely the pairwise correlation calculations (Lemma 2) and the choice of adversary, comes directly from [DKS17]. Similarly, the upper bound for $L_q$-sparse mean testing is a minor modification of the proof in [Li17].

*Assessment*: I would recommend acceptance because of the comprehensive nature of the result: while dense mean testing was already understood, it is quite cool that they managed to demonstrate that the phase transition for non-robust mean testing disappears altogether.

---

> ### Author Response · Authors · 2022-08-01
> **Respose to the review by PDuZ**
>
> We thank the reviewer for their time and thoughtful comments. We address the specific questions below.
>
> **Response to Questions**
>
> > Do the lower bound techniques get the right $\gamma$  scaling as well? If not, where are the bottlenecks in the argument that preclude this?
>
> We thank the reviewer for bringing up this important point. For the lower bounds in Theorems 4 and 5, the dependence on $\varepsilon$ and $\gamma$ is $\Omega(\varepsilon ^4/\gamma^4)$. The exact dependence on these parameters is still an open problem even for robust Gaussian mean testing (non-sparse). We conjecture that the actual dependence on these parameters should scale as $\varepsilon ^2/\gamma^4$, and hence we believe that our dependence on these parameters is not tight (since we have $\varepsilon = o(\gamma)$). In the current proof, the bottleneck appears while obtaining an upper bound for the chi-squared correlation between two Huber-contaminated distributions, but we suspect that more advanced techniques (like Moment Matching, for higher moments) might be required to get a tight dependence on $\gamma$ and $\varepsilon$.  We will include these points as a remark in the final version of the paper.
>
> > Is Brennan-Bresler's lower bound for robust SLR also a lower bound against robust SLR testing?
>
> We believe that this is the case in the sparse regime ($s=o(\sqrt{d})$). However, we have not gone through their arguments rigorously, and will confirm this before finalizing the paper.
>
> > Minor typos
>
> We are grateful to the reviewer for bringing these typos to our attention. We will correct these in the final version of the paper.

---

> > ### Comment · Reviewer_PDuZ · 2022-08-08
> > **Thanks!**
> >
> > I'm happy with the author response and will maintain my score.

---

### Official Review · Reviewer_8QBu · 2022-07-11

**Rating:** 7
**Confidence:** 4
**Soundness:** 3 good
**Presentation:** 3 good
**Contribution:** 3 good

**Summary:**

The paper studies the sample complexity of robust sparse Gaussian mean testing and linear regression. Motivated by prior work saying that the number of samples required for testing versions of various inference tasks becomes as high as that required for the learning version of the problems (i.e., the sample complexity experiences an information-theoretic gap when corruptions are introduced), the paper asks whether the same is true in the sparse setting. For the robust $s$-sparse Gaussian mean testing problem, it shows a lower bound of $\Omega(s \log(d/s))$ for the sample complexity, which means that, in contrast to the non-robust setting, the sample complexity does not default to $\sqrt{d}$ when $s \geq \sqrt{d}$. The result extends to a more general notion of sparsity, for which the paper also obtains upper bounds on the sample complexity. Finally, they show qualitatively similar results for the sparse linear regression setting (in the usual notion of sparsity). The approach for the mean testing result involves defining a family of $\epsilon$-corrupted Gaussians (in the Huber’s contamination model) with large mean  and considering the binary hypothesis testing problem that asks for distinguishing this family from the standard Gaussian (i.e., zero-mean). The proof is based on Le Cam's method and utilizes a lemma of [DKS17] to bound the chi-square distance between the standard Gaussian and the uniform mixture over the aforementioned family. The linear-regression result has to take care of the additional complication regarding some low-probability events that cause the chi-square divergence to blow up.

**Questions:**

Is it entirely accurate to say that the testing is ``much harder than its non-robust counterpart'',  given that the sample complexity agree in the very sparse regime $s < \sqrt{d}$ ?

**Limitations:**

.

**Strengths And Weaknesses:**

Significance: Testing means of Gaussians and linear regressors are one of the most fundamental problems in statistics and this work completes the characterization of the sample complexity for the robust version.

Quality: The arguments seem to be overall correct. To the extent that I checked the proofs in the supplementary material, I do not have issues regarding soundness.

Clarity: The paper is well-written. The sketch in Section 1 is clear in conveying the high-level approach.

Originality: The techniques for proving the lower bounds are based on Le Cam’s method and [DKS17]. The definition of the hard distribution class is somewhat natural, and the approach does not need to significantly deviate from the prior established technology. The linear regression result however requires more substantial work at the technical level. The upper bound in Theorem 7 follows by modification of standard arguments.

The paper fills the gap in our understanding regarding the sample complexity for robust sparse testing. Although the problems do not require substantial originality in terms of new techniques, the solution is non-trivial and the results are of interest to the robust statistics community.

---

> ### Author Response · Authors · 2022-08-01
> **Response to the review by 8QBu**
>
> We thank the reviewer for their time and feedback. We also thank the reviewer for going through the proofs in the paper. We address the specific question below.
>
> **Response to Questions**
>
> > Is it entirely accurate to say that the testing is ``much harder than its non-robust counterpart'', given that the sample complexity agree in the very sparse regime $s<\sqrt{d}$ ?
>
> We thank the reviewer for raising this important aspect. Indeed, testing becomes much harder only in the dense regime ($s>\sqrt{d}$). In the sparse regime ($s<\sqrt{d}$), the difference in sample complexities between the non-robust and robust cases is at most logarithmic. We will clarify the quoted statement in the final version of the paper.

---

### Official Review · Reviewer_jAix · 2022-07-12

**Rating:** 6
**Confidence:** 3
**Soundness:** 4 excellent
**Presentation:** 4 excellent
**Contribution:** 2 fair

**Summary:**

The paper focuses on two problems within the domain of robust statistics:
1. Sparse Gaussian mean testing: the goal is to robustly testing whether the mean of a Gaussian is zero or far from zero when we are guaranteed that the mean vector $\theta$ satisfies some form of sparsity codified as $\|\theta\|_q = s$ for some $q \in [0,2)$.
2. Testing in the sparse linear regression model: given $x \sim \mathcal{N}(0,I_d)$ and a sparse $\theta$ we have $y = x^\top \theta + \epsilon_i$ and our goal is to test whether $\|\theta\|_2 = 0$ or $\|\theta\|_2 \ge \gamma$.
In both problems we have that an $\epsilon$ fraction of samples are arbitrarily corrupted by an adversary.

The main results of the paper are information theoretic lower bounds for the above tasks which depend on $s,d$ (the authors focus on the regime where $\gamma,\epsilon$ are constants and the dependence of the bounds on these quantities is not focused upon).
The non-robust versions of these questions show a dependence of the following form (when $q=0$):
1. $\Theta\left(s\log(1+d/s^2)\right)$ if $s < \sqrt{d}$,
2. $\Theta\left(\sqrt{d} \right)$ if $s \ge \sqrt{d}$.
In particular, the testing version of the problem can always be solved more efficiently than the learning/estimation version. However, in the robust variant of these problems the results of this paper show that this is no longer the case. Irrespective of $s$, the robust sparse Gaussian mean testing problem requires $\Omega(s\log(d/s))$ samples.

In addition to the $q=0$ setting, the authors also provide upper and lower bounds to these problems to the $q \in (0,2)$ settings.
The main message of the story remains quite similar in the sparse linear regression testing problem as well.
Among the techniques used by the authors are the well-known Le Cam's two point method, some inequalities connecting different distance metrics to each other, and some inequalities which help bound the $\chi^2$ distance of a product of a mixture of distributions with another product distribution.

**Questions:**

**Questions:**
1. What are the dependences of the bounds on $\epsilon,\gamma$ in Theorems 4 and 5? How tight are these?
2. It would be nice to see some discussion around the setting of non identity covariance matrices and testing between two Gaussians with different covariances.
3. The sparse testing problem exhibits a phase transition in its sample complexity as a function of sparsity. Namely, we only see benefits of sparsity until the sparsity parameter reacher $\sqrt{d}$. It is intriguing that a similar phase transitionary behavior doesn't occur here and the situation seems to linearly degrade with $s$ for all ranges of $s$. Is there any high-level intuition on what causes the phase transitionary behavior in the non-robust setting and why this is no longer the case in the robust setting?

**Suggestions:**
1. "Upper bounds of [Li17] and Theorem 2 are computationally inefficient...” – present the computational complexity in terms of $d, s, \gamma$ here.
2. Line 138 of supplementary typo:  “note that for in order to”

**Limitations:**

The authors adequately addressed the potential negative societal impact of their work. They also adequately address the limitations of their results.

**Strengths And Weaknesses:**

**Strengths:**
1. Sparsity and robustness are two important sub-fields within Statistics research today which the paper focuses on.
2. The paper's contributions are original. In addition, the paper is well written and clear to read.
3. The authors provide a thorough and complete theoretical analysis of the problem leading to a tight understanding of the sample complexity along with an extensive survey of all related work.


**Weaknesses:**
1. Even within the scope of robustly testing a sparse Gaussian the paper focuses on a specific niche problem of mean testing. It would be nice to see some discussion around the setting of non identity covariance matrices and testing between two Gaussians with different covariances.
2. The paper presents a nice overview of the techniques used and has a number of results. But it is not fully clear where the main technical challenges arise in this setting. In particular, it is hard for me to understand why the problems in the robust setting do not exhibit a phase transitionary behavior in the sample complexity. Given this, I am a little unsure on the significance of the results of the current paper.

---

> ### Author Response · Authors · 2022-08-01
> **Response to the review by jAix**
>
> We thank the reviewer for their time and thoughtful comments. We address their specific questions below.
>
> **Response to Questions**
>
> 1. We thank the reviewer for bringing up this important point. For the lower bounds in Theorems 4 and 5, the dependence our proof yields with respect to $\varepsilon$ and $\gamma$ is $\Omega(\varepsilon^4/\gamma^4)$. Importantly, the exact dependence on these parameters is still an open problem even for “dense” robust Gaussian mean testing (non-sparse). We conjecture that the actual dependence on these parameters is $\varepsilon^2/\gamma^4$, and hence we believe that our dependence on these parameters is not tight (since we have $\varepsilon = o(\gamma)$). However, this is not just an artefact of our analysis: our current techniques (as well as the ones from previous work on the "dense" robust testing case) cannot yield better than an $\Omega(\varepsilon ^4/\gamma^4)$ dependence (we refer the reviewer to the response to reviewer PDuZ for additional details). We will elaborate on these points in the final version of the paper.
>
> 2. Thank you for this query. We agree that handling the case of arbitrary covariances is a natural and important question; however, we believe that deriving the sample complexity in the case of non-identity (and unknown) covariance matrices would require significant additional effort, and new techniques and ideas, and thus we left it for future work.
>
>     It is worth pointing out that we are not aware of any work addressing the sample complexity results for non-identity covariances even in the non-robust (but sparse) setting. The closest which comes to mind is the problem of robustly testing the covariance of a Gaussian, as discussed in [DK21].  We will try to include a discussion regarding this in the final version.
>
> 3. We believe that the presence of phase transition in the non-robust setting is due to the dominance of two different behaviors based on the value of sparsity parameter $s$. On the one hand, for an $s$-sparse signal, if we knew the non-zero coordinates a priori, then the testing problem is no different from the testing of a $s$-dimensional signal. So, the sample complexity would be $O(\sqrt{s})$. However, the number of samples required to detect the non-zero coordinates scales as $\tilde{O}(s)$ (where $\tilde{O}$ hides polylogarithmic factors in the argument). This is because the magnitude of each coordinate scales (in the worst case) as $O(\gamma/\sqrt{s})$ and therefore requires $O(s)$ samples to achieve the same Signal to Noise Ratio after averaging. So this strategy leads to an $\tilde{O}(\sqrt{s}+s) = \tilde{O}(s)$ sample complexity. On the other hand, there is always the option of ignoring the sparsity altogether and performing the testing of the $d$-dimensional mean in that case has sample complexity $O(\sqrt{d})$. Combining the two leads to a non-robust mean testing sample complexity of $\tilde{O}(\min(s,\sqrt{d}))$, leading to the two regimes, with a transition at $s=\sqrt{d}$. As it turns out, this simple two-pronged approach is actually optimal in the non-robust case.
>
>     However, the phase transition vanishes in the robust case because we can always match the first moment of the input distribution (a Gaussian with non-zero mean vector) to that of a zero-mean Gaussian by choosing a suitable adversarial distribution. Hence we will have to rely on higher moments for testing, which increases the sample complexity: there is no longer an $O(\sqrt{d})$ upper bound option, as there was in the non-robust case.
>
> **Response to Suggestions**
>
> 1. Thank you for this suggestion. The algorithm has exponential time complexity in $s$ and polynomial in $d$ and $\gamma$. We will add this clarification in the final version.
>
> 2. Thank you for pointing out this typo. We will correct it in the final paper.

---

> > ### Comment · Reviewer_jAix · 2022-08-06
> > **Thank you for the response**
> >
> > Your response addresses most of my questions and helps my assessment of the significance of your results.
> > Due to the overlap in the techniques required to prove the current lower bounds with lower bounds in the non-sparse setting, I will maintain my score.

---

### Meta-Review · Area_Chair_6vxg · 2022-08-26

**Recommendation:** Accept
**Confidence:** Certain

**Metareview:**

The reviewers and I agree that this result is a solid, if not groundbreaking, result in the theory of robust statistics. It considers a very natural testing problem, and when the fraction of corrupted samples is constant, completely settles the statistical complexity of the problem. There are some concerns that its immediate practical impact is limited, but overall, the consensus is that the paper represents a solid technical contribution, and will be of interest to the robust statistics community, and the learning theory community at large. Therefore, we believe this paper is above the bar for acceptance at NeurIPS.

**Award:**

No

---

### Decision · Program_Chairs · 2022-09-14

Accept